# 3D-Prover: Diversity Driven Theorem Proving With Determinantal Point Processes

## Abstract

A key challenge in automated formal reasoning is the intractable search space, which grows exponentially with the depth of the proof. This branching is caused by the large number of candidate proof tactics which can be applied to a given goal. Nonetheless, many of these tactics are semantically similar or lead to an execution error, wasting valuable resources in both cases. We address the problem of effectively pruning this search, using only synthetic data generated from previous proof attempts. We first demonstrate that it is possible to generate semantically aware tactic representations which capture the effect on the proving environment, likelihood of success, and execution time. We then propose a novel filtering mechanism which leverages these representations to select semantically diverse and high quality tactics, using Determinantal Point Processes. Our approach, 3D-Prover, is designed to be general, and to augment any underlying tactic generator. We demonstrate the effectiveness of 3D-Prover on the miniF2F-valid and miniF2F-test benchmarks by augmenting the ReProver LLM. We show that our approach leads to an increase in the overall proof rate, as well as a significant improvement in the tactic success rate, execution time and diversity.

## 1 Introduction

Interactive Theorem Proving, as the name suggests, has traditionally involved a human guiding a proving system to verify a formal proposition. It has found applications in a wide range of fields, from secure software (Tan et al., 2019) to the verification of mathematical results (Hales et al., 2017). There has been significant interest in automating this process, with formalization efforts requiring a high level of human expertise (Klein et al., 2009). Beyond this, it is considered a 'grand challenge' for AI, requiring a high level of reasoning and planning to be successful (Reddy, 1988). Even the largest current models struggle with the complexity of the task, with for example GPT-4 only able to solve 13.5% (Thakur et al., 2023) of the high school level miniF2F-test (Zheng et al., 2021) benchmark. This has motivated the development of specialized models and search algorithms to address the unique challenges of the domain (Wang et al., 2024; Polu et al., 2022; Jiang et al., 2022b; Han et al., 2022; First et al., 2023; Zhao et al., 2023; Wang et al., 2023; Whalen, 2016; Wu et al., 2021b; Wang et al., 2018; Wang & Deng, 2020; Rabe et al., 2020; Polu & Sutskever, 2020; Mikuła et al., 2023; Loos et al., 2017; Li et al., 2021; Lewkowycz et al., 2022; Jiang et al., 2021; 2022a; Gauthier et al., 2017).

With most non-trivial proofs requiring long chains of correct reasoning, it is a challenge to generate them in one pass without mistakes. The addition of a search algorithm is common for addressing this, as is done by the current state-of-the-art DeepSeek-Prover-V1.5 (Xin et al., 2024). Under this paradigm, candidate tactics are generated and executed in the proving system, which (if successful) results in new subgoals to prove. This generates a tree of possible proof paths, where a search algorithm selects the most promising nodes to expand. The primary challenge faced by these approaches is the exponential growth in the number of proof paths, limiting the complexity of the problems that can be solved efficiently.

Many of the generated tactics are equivalent, modulo variable renaming and other semantics-preserving transformations. See Figure 1 for a sample search tree from the ReProver (Yang et al., 2023) system, where several semantically similar paths are explored, wasting valuable resources. Simple lexical similarity scores fail to cover the semantics (meaning) of a tactic, as captured by the

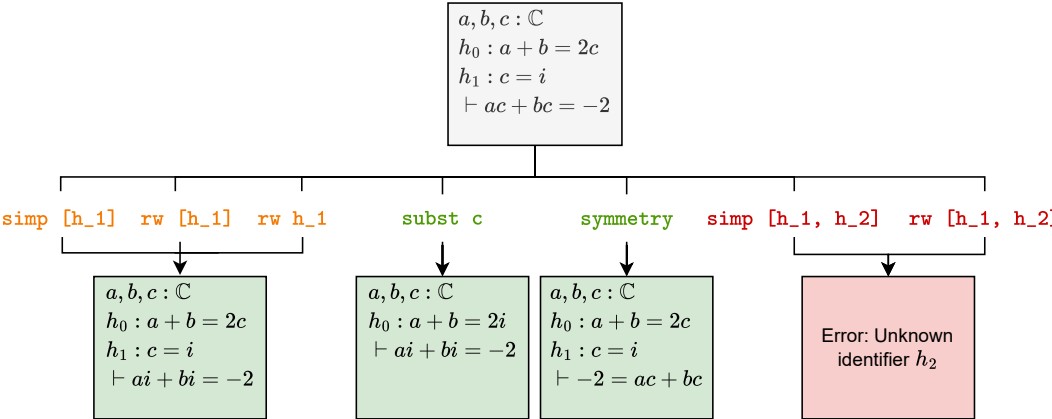

Figure 1: An example node expansion for a failed ReProver attempt, which our DPP model was able to prove. Tactics on the left result in the same proof state, tactics on the right result in an error, and tactics in the centre result in a unique proof state. The high error rate and tactic similarity motivates our filtering approach, which prunes the search space to give a diverse set of subgoals.

effect of the tactic on the environment. For example, an expression and its negation vary by only a single character, but have a large semantic difference. It is therefore desirable to filter tactics by their semantic rather than syntactic diversity. In addition, many tactics lead to an execution error from the prover. From our experiments with miniF2F, we find approximately 75% of tactics result in an execution error (Section 2.2). With the execution of tactics in the environment being expensive, this further restricts the space of proofs which can be explored efficiently.

These challenges motivate our proposed approach, **D**iversity **D**riven **D**eterminantal **P**oint **P**rocess **P**rover (3D-Prover). 3D-Prover adds an extra 'dimension' to existing proving systems by including a filtering mechanism on top of the existing tactic generation and search components. 3D-Prover uses Determinantal Point Processes (Kulesza, 2012) to prune the proof search space by filtering tactic candidates to a diverse and high quality subset. The large amount of synthetic data generated from proof attempts enables us to learn the effect tactics have on the environment, including the likelihood of an error and the execution time. We leverage this to generate tactic representations which reflect their semantics, which 3D-Prover uses to filter tactics based on a combination of their diversity and quality. 3D-Prover allows for a direct tradeoff between search objectives, with hyperparameters controlling the weighting of error, time and diversity in the filtering process. 3D-Prover is a general approach which can be used to augment any underlying tactic generator. We demonstrate this by augmenting the open source ReProver LLM to obtain a significant improvement in the success rate, execution time and diversity of tactics, and the overall proof success rate.

We summarize our contributions as follows:

- We study the feasibility of learning the environment dynamics of proving systems. We demonstrate tactic representations which capture the likely effect on the environment, using them to predict the likelihood of success and execution time of a tactic, as well as the resulting proof state or error message.
- We propose a novel edge filtering approach using Determinantal Point Processes (Kulesza & Taskar, 2011), which leverage these representations to select semantically diverse subsets of quality tactics. Our method is modular and can be used with any underlying tactic model.
- We evaluate our approach by augmenting ReProver (Yang et al., 2023) on the miniF2F (Zheng et al., 2021) benchmark, where we demonstrate a significant improvement in the tactic success rate, diversity and overall proof success rate.

## 1.1 RELATED WORK

There is little prior work on learning the effect of a tactic on the proving environment. Xin et al. (2024) recently included successful environment responses as an auxiliary learning objective, how-

ever do not investigate the task in detail. We extend this by modeling the error likelihood, error messages and execution time, which we use to generate useful tactic representations. Several approaches have used previous proof attempts to improve performance, using the sparse binary signal from the proof status of a goal (Bansal et al., 2019; Wu et al., 2021a). This has been used to improve search algorithms, as done in (Lample et al., 2022; Polu et al., 2022; Wang et al., 2023). These approaches do not consider the diversity of the nodes expanded, with nothing preventing the search from exploring semantically similar paths. Xin et al. (2024) uses intrinsic reward for exploration by rewarding new nodes in the search tree. The addition of any node is rewarded equally, even if they are similar (but not identical) to existing nodes. We instead select tactics based on their diversity with respect to the resulting nodes. First & Brun (2022) use a diverse ensemble of models to improve proof performance, whereas we focus on diversity with respect to the environment response, given an arbitrary underlying model (or models).

## 1.2 BACKGROUND: DETERMINANTAL POINT PROCESSES

Determinantal Point Processes (DPPs) are a class of probabilistic models for sampling subsets from a ground set $\mathcal{Y}$. They provide an inherent trade-off between the diversity and quality of the sampled subsets, successfully being applied to this end across a variety of domains (Kulesza, 2012; Hsiao & Grauman, 2018; Zhang et al., 2016). This motivates their use in our filtering approach (Section 3.2).

In line with Kulesza (2012), for $|\mathcal{Y}| = n$ we define the kernel $L \in \mathbb{R}^{n \times n}$ of a DPP as the Gram matrix $L = B^T B$ for $B \in \mathbb{R}^{n \times d}$, where column $\boldsymbol{b}_i \in \mathbb{R}^d$ of $B$ is a vector representing element $i \in \{1, \ldots, n\}$ of $\mathcal{Y}$. These vectors $\boldsymbol{b}_i$ are commonly decomposed into a set of unit norm diversity features $\boldsymbol{\phi}_i \in \mathbb{R}^d$ and quality scores $q_i \in \mathbb{R}^+$, so that $\boldsymbol{b}_i = q_i \boldsymbol{\phi}_i$, $||\boldsymbol{\phi}_i|| = 1$ for all $i \in \{1, \ldots, n\}$. The similarity matrix $S$ is then defined as $S_{ij} = \boldsymbol{\phi}_i^T \boldsymbol{\phi}_j$. The probability of sampling a subset $A \subseteq \mathcal{Y}$ from a DPP is then proportional to the determinant of the submatrix of $L$ indexed by $A$, $\mathbb{P}(A) \propto \det(L_A) = (\prod_{i \in A} q_i^2) \det(S_A)$. Geometrically, this determinant is the volume of the parallelepiped spanned by the submatrix $L_A$, which as we see in Figure 3, is maximised based on a combination of the similarity and length (quality) of the chosen elements. In this way, DPPs elegantly trade off between the quality and diversity of elements. Normally the size of the sampled subset $|A|$ is variable, however Kulesza & Taskar (2011) introduce $k$-DPPs which restricts the size of the subset to a fixed $k \in \mathbb{N}$, and where the probability of sampling $A$ is normalised over subsets of size $k$. That is, for a $k$-DPP, $\mathbb{P}(A) = \det(L_A) / \sum_{|A'|=k} \det(L_{A'})$.

## 2 TRANSITION AWARE REPRESENTATION LEARNING

One proof attempt can generate a rather large amount of data. A single pass of the miniF2F-valid benchmark of 244 proofs results in approximately 500,000 transitions, capturing rich information about the error likelihood, execution time and resulting proof state or error message. This section explores the feasibility of using this data to learn how tactics affect the environment. We operationalise this as a supervised learning task: given a goal and tactic, we predict the error status, execution time and environment output. We effectively learn these targets from only this synthetic data, and embed this information into a compact tactic representation. The upshot, as we show in Section 3, is that these representations can be utilised to improve the performance of subsequent proof attempts.

### 2.1 TRANSITION MODELS

The result of a proof attempt (formalised in A.1) is the dataset $\mathcal{D}$ of *transitions* $\{(g, t, s, \tau, o)\}$, which captures the results of applying *tactics* $t \in \mathcal{T}$ to *goals* $g \in \mathcal{S}$. The *status* $s \in \{0, 1\}$, indicates a success (1) or failure (0), $\tau \in \mathbb{R}$ gives the execution time of the tactic and the *output* $o \in \mathcal{O}$ is the environment response, which is an error message, new goals to prove, or a proof success. We propose a method of learning tactic representations $\boldsymbol{e} \in \mathbb{R}^d$ which capture the result $(s, \tau, o)$ of applying $t$ to $g$. By using these as diversity features for DPP, we can filter tactics based on the diversity of their outcomes, before they are executed.

We define our *transition model* $\xi : \mathcal{S} \times \mathcal{T} \to [0, 1] \times \mathbb{R} \times \mathcal{O}$ as a mapping from a goal $g$ and tactic $t$ to an estimate of the status $s$, time $\tau$ and output $o$. To ensure $\xi$ admits effective representations in $\boldsymbol{e}$, we construct it with three components. The Encoder $E : \mathcal{S} \times \mathcal{T} \to \mathbb{R}^d$ takes the goal $g$ and tactic $t$ as input, and outputs our representation $E(g, t) = \boldsymbol{e}$. As $\boldsymbol{e}$ will be used as the diversity feature for

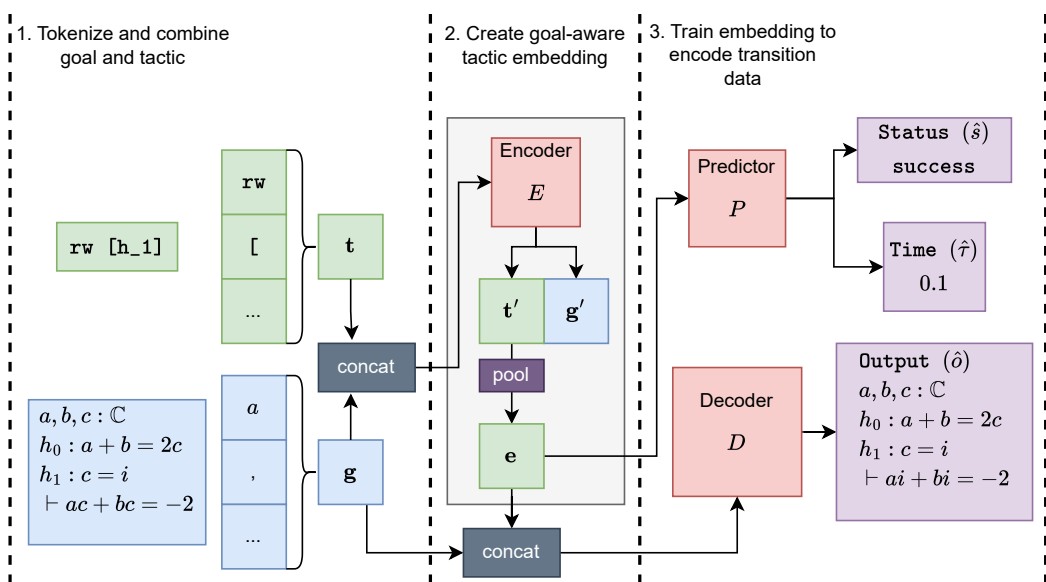

Figure 2: Our COMBINED architecture for learning transition aware tactic embeddings. The tokenized tactic $\mathbf{t}$ and goal $\mathbf{g}$ are concatenated and passed through the Encoder $E$. A representation vector $\mathbf{e}$ is generated by mean-pooling over the tactic token embeddings $\mathbf{t}'$. The Predictor $P$ takes this embedding and predicts whether the tactic results in an error (Status), and the execution time (Time). The Decoder $D$ takes the embedding and goal to predict the environment response (Output), which is either an error message or new goals to prove. The result is a compact representation of the tactic which captures its effect on the proving environment, enabling our proposed filtering model.

DPP, it is constrained to have unit norm $||\boldsymbol{e}|| = 1$. The Predictor $P : \mathbb{R}^d \to [0, 1] \times \mathbb{R}$ maps $\boldsymbol{e}$ to an error probability for the status and a score for the time prediction, with $P(\boldsymbol{e}) = (\hat{s}, \hat{\tau})$. The Decoder $D : \mathbb{R}^d \times \mathcal{S} \to \mathcal{O}$ maps $\boldsymbol{e}$ and $g$ to the output prediction, such that $D(\boldsymbol{e}, g) = \hat{o}$. The transition model is then

$$\xi(g, t) = (P(E(g, t)), D(E(g, t), g)) = (P(\boldsymbol{e}), D(\boldsymbol{e}, g)) = (\hat{s}, \hat{\tau}, \hat{o}). \tag{1}$$

We note that the Decoder and Predictor can only access information of $t$ through $\boldsymbol{e}$. Hence our architecture requires the Encoder to learn an effective representation for $\boldsymbol{e}$, so that the Decoder and Predictor can use this to determine the subsequent effect of the tactic on the environment.

## 2.2 EXPERIMENTS

For our experiments, we use an Encoder-Decoder Transformer for the Decoder $D$, and an Encoder-Only Transformer for the Encoder $E$. We take the pretrained ReProver (Yang et al., 2023) LLM to initialise both components. We implement the Predictor $P$ as a single hidden layer MLP, with hidden dimension $d/2$ (where $d = 1472$) and two real valued output nodes. The time prediction $\hat{\tau}$ is the output of the first node, and the status prediction $\hat{s}$ is taken as the sigmoid of the second. We use this simple Predictor architecture to speed up our filtering algorithm presented in Section 3

We investigate four variations of the transition model $\xi$. For the COMBINED model (Figure 2), the tactic is concatenated with the goal, and the embeddings from the Encoder are computed for all tokens. We then generate a single tactic embedding by mean-pooling over the tactic tokens. We compare this with the SEPARATE model which encodes the tactic without attending to the goal. We hypothesise that allowing the tactic tokens to attend to the goal will allow the Encoder to better represent the semantics of the tactic. To form a naive baseline, we implement a NO TACTIC model which does not use the tactic at all, and instead uses only the goal tokens. We do this to account for any inherent patterns in the goal which may be predictive of the outcome, for example a particular goal which has a high error rate. This allows us to ground our results in the performance of this baseline, so we can observe the direct effect of the tactic in predictive performance. We also

compare with an **ALL TOKENS** model which uses all tactic tokens for the Decoder without reducing to a single embedding. We maintain the pooling operation over the tactic tokens for the status and time prediction tasks, but allow the Decoder to attend to all tokens for the output prediction. We implement this comparison to see the degree of information loss induced by reducing tactics to a single vector.

Given $\alpha_s, \alpha_\tau, \alpha_o \in \mathbb{R}^+$, with estimates $\hat{s}, \hat{\tau}, \hat{o}$ and for a minibatch $\mathcal{B} \subseteq \mathcal{D}$, we optimise the transition loss

$$\mathcal{L}_\mathcal{T}(\mathcal{D}, \xi) = \sum_{(g,t,s,\tau,o) \in \mathcal{B}} \alpha_s \mathcal{L}_s(s, \hat{s}) + \alpha_\tau \mathcal{L}_\tau(\tau, \hat{\tau}) + \alpha_o \mathcal{L}_o(o, \hat{o}). \tag{2}$$

The hyperparameters $\alpha_s, \alpha_\tau, \alpha_o$ control the weighting of the status, time and output losses. For simplicity, we set these to 1, however they could be tuned to reflect the relative importance of each task. We use the binary cross-entropy loss $\mathcal{L}_s$ for the status prediction, the mean squared error (MSE) $\mathcal{L}_\tau$ for the time prediction, and the cross-entropy loss $\mathcal{L}_o$ for the output prediction.

We obtain the dataset $\mathcal{D}$ from a vanilla ReProver attempt on miniF2F-valid, which results in 498,236 transitions, which we split randomly into 95% training, 5% testing. There is the possibility of dependence between the splits, as the test set includes goals seen in training with different tactics. The NO TACTIC baseline should capture any of this, with our results in Section 3.2.1 showing our representations generalise from miniF2F-valid to miniF2F-test. For the error prediction task, we reweight classes to account for imbalance, which is approximately 75% error, 25% success. We use the AdamW optimizer, with a learning rate of $10^{-5}$ and a batch size of 1. We train each model for 2 epochs on a single RTX4090, and report the results on the test set.

### 2.2.1 RESULTS

| Embedding | Output | | | Status | | | Time |
|---|---|---|---|---|---|---|---|
| | BLEU | ROUGE-L F1 | Top-4 | F1 | TPR | TNR | MSE |
| ALL TOKENS | 0.31 | 0.38 | 0.31 | 0.85 | 0.82 | 0.96 | 0.17 |
| COMBINED | **0.33** | **0.39** | **0.32** | **0.88** | **0.85** | **0.97** | **0.16** |
| SEPARATE | 0.27 | 0.34 | 0.27 | 0.76 | 0.71 | 0.94 | 0.28 |
| NO TACTIC | 0.17 | 0.22 | 0.13 | 0.22 | 0.14 | 0.96 | 0.37 |

Table 1: Results for predicting unseen environment responses given a goal and tactic, for transitions from miniF2F-valid. The NO TACTIC result forms a baseline to assess the impact of the tactic representation. We observe that any tactic representation enables far better predictions, and constraining these to a single vector (COMBINED and SEPARATE) does not hurt the performance gain. This demonstrates tactic representations which capture their effect on the environment, enabling our filtering model in Section 3. Comparing the COMBINED and SEPARATE models, allowing the representation to attend to the goal leads to a large improvement.

To assess the Output prediction, we use beam search to generate 4 candidate outputs for each transition. We use the BLEU (Papineni et al., 2002) and ROUGE-L (Lin, 2004) scores to assess the quality of the highest scoring beam in comparison to the ground truth, which is either an error message or a new set of subgoals. We also report the Top-4 accuracy, which is the proportion of samples which have one beam identical to the ground truth. For the Status prediction task, we take the prediction as 1 if $\hat{s}_{ki} > 0.5$ and 0 otherwise, reporting the F1 score, true positive rate (TPR) and true negative rate (TNR). The Time MSE is the mean squared error of the time prediction over the test set.

Table 1 summarises the performance of our transition models on the test set. Our results suggest tactic representations which capture useful information about their effect on the environment, which we can see by the clear improvement across all approaches compared to the NO TACTIC baseline. The higher scores across all metrics of the COMBINED versus the SEPARATE model support our hypothesis that we can better predict transitions when the tactic embedding attends to the goal. The ALL TOKENS model, where we allow the Decoder to attend to the full tactic, does not increase performance in comparison to the COMBINED model. This shows that we can effectively represent the tactic as a single embedding without any loss of information. Our results demonstrate the feasibility

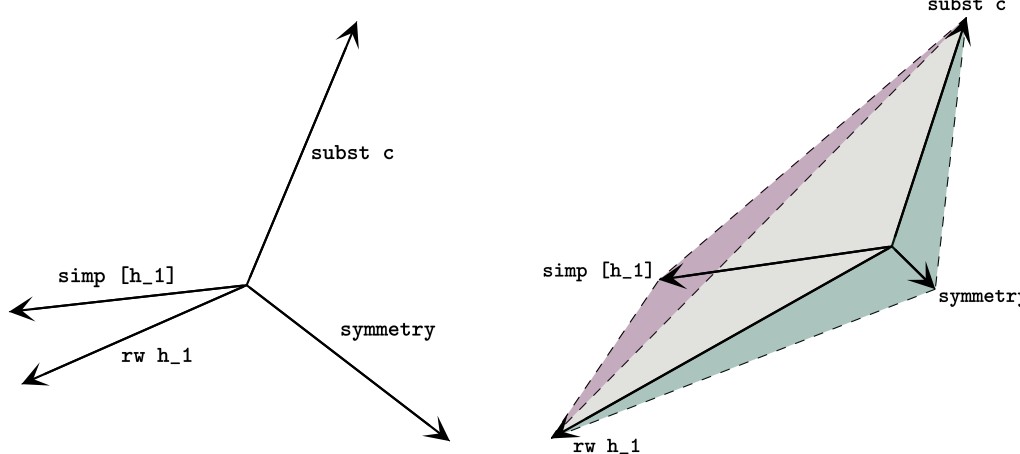

(a) Initial unit norm tactic embeddings $\phi_i$, representing the predicted environment response.

(b) Embeddings scaled by quality ($q_i$), giving vectors $q_i \phi_i$ to be filtered by $k$-DPP

Figure 3: DPP for tactic filtering. The tactic embeddings from the transition model are scaled by quality scores, before a subset of tactics are selected using $k$-DPP. Subsets are chosen proportionally to the area spanned by their elements, giving a combination of quality and diversity. For this simplified example, we take the 2D PCA projection of embeddings for tactics in Figure 1, setting the quality to the scaled generator logits. Comparing the shaded areas in (b) and assuming `subst c` and `rw` $h_1$ have been selected, we see that `symmetry` is favoured over `simp` [$h_1$]. Although `simp` [$h_1$] is scored higher by the generator, it is less diverse with respect to `subst c` and `rw` $h_1$.

of learning the environment dynamics of proving systems. To illustrate the difficulty of this task, we include all prediction examples for the COMBINED model, along with their ground truth, in the supplementary material.

## 3 FILTERING MODEL

In the previous section we used synthetic proof data to generate semantically aware tactic representations, allowing us to predict the likelihood of success, execution time and environment response. We now take this a step further by using these representations to augment proof search. We begin by introducing our filtering model 3D-Prover, which prunes tactic candidates based on their quality and the semantic diversity of their representations. We show that 3D-Prover is able to improve the performance of the ReProver LLM on the miniF2F-valid and miniF2F-test benchmarks, particularly when a deeper search configuration is used. We conclude with a multifaceted ablation study showing the effect of our filtering model on the success rate, number of unique responses and execution time.

### 3.1 FILTERING MODEL

Algorithm 1 defines our filtering model, 3D-Prover, which maps a list of tactics $T$ from the underlying tactic policy $\pi_0$ to a subset $T'$ of size $K$. We use the Encoder $E$ and Predictor $P$ defined in Section A.1 to generate unit norm tactic embeddings $\phi_i$ and predict the time and error likelihood. The embeddings $\phi_i$ encode the predicted environment response through their direction only, as they are unit norm (Figure 3). The quality scores $q_i$ then scale these tactics based on the underlying model logits $m_i$, as well as the predicted error likelihood $s_i$ and execution time $\tau_i$. We have hyperparameters for the normalisation temperature $\theta$, as well as the error and time weights $\lambda_s, \lambda_\tau$. The parameter $\theta$ controls the scaling temperature of the model logits, with a higher temperature flattening out the distribution. It therefore adjusts the diversity bias of the filtering model by reducing the impact of the quality scores when sampling. We then compute the kernel $L$ from $q_i$ and $\phi_i$, and sample a subset of tactics $T'$ using the $k$-DPP algorithm (Kulesza & Taskar, 2011).

---

**Algorithm 1:** 3D-Prover

---

**Input** : Goal $g$, candidate tactics $T = \{t_i\}_{i=1}^N$, filter size $K$, Encoder $E$, Predictor $P$, error
weight $\lambda_s$, time weight $\lambda_\tau$, temperature $\theta$, underlying tactic policy $\pi_0$
**Output:** Filtered tactics $T' \subset T$

    // Compute embeddings and scores for each tactic
1 **for** $i$ *in* $\{1, \ldots, N\}$ **do**
2     $\phi_i \leftarrow E(g, t_i)$ ;                  // Compute tactic embedding
3     $(s_i, \tau_i) \leftarrow P(\phi_i)$ ;          // Compute time and error scores
4     $\tau_i \leftarrow 1 - \frac{\tau_i}{||\boldsymbol{\tau}||}, \boldsymbol{\tau} = (\tau_1, .., \tau_N)$ ;    // Normalise time scores
5     $m_i \leftarrow \frac{\exp(\pi_0(t_i|g)/\theta)}{\sum_{j=1}^N \exp(\pi_0(t_j|g)/\theta)}$ ;    // Normalise model logits
6     $q_i \leftarrow m_i + \lambda_s s_i + \lambda_\tau \tau_i$ ;        // Compute quality score
    // Filter tactics with $k$-DPP
7 $L \leftarrow B^T B$, where $B = [q_1\phi_1, \ldots, q_N\phi_N]$ ;    // Compute kernel matrix
8 Compute eigenvalues $\lambda_i$ and eigenvectors $\boldsymbol{v}_i$ of $L$
9 Sample $J \subset \{1, \ldots, N\}$ using Algorithm 2 of Kulesza & Taskar (2011),
10   with parameters $\{(\boldsymbol{v}_i, \lambda_i)\}$, $k = K$
11 **return** $T' = \{t_j\}_{j \in J}$

---

## 3.2 EXPERIMENTS

To test the performance of 3D-Prover, we use ReProver (Yang et al., 2023) as the underlying tactic policy $\pi_0$, with the Encoder $E$ and Predictor $P$ components as defined in Section 2. We chose ReProver as it is a small ($\sim$ 300M parameters), popular and performant open source model, allowing us to run our experiments in a reasonable timeframe. We run our experiments in Lean 3 (De Moura et al., 2015) using the BAIT (Lamont et al., 2024) platform with a modified LeanDojo (Yang et al., 2023) environment, where we set an environment timeout of 600 seconds per proof attempt. We train a combined transition model on the miniF2F-valid benchmark, and use the Encoder and Predictor components to generate tactic embeddings and quality scores as per Algorithm 1. We first examine the performance of 3D-Prover without any hyperparameter tuning, setting $\lambda_s = \lambda_\tau = 0$, $\theta = 1$. We then perform ablation studies using miniF2F-valid to examine the influence of the hyperparameters on the tactic success rate, execution time and diversity of the environment response. For miniF2F-test, we allow the model four attempts per proof to increase confidence in the results, while for miniF2F-valid we allow one attempt per configuration to facilitate a wider set of ablations.

We set the search policy for all experiments to be Best First Search (BestFS), where nodes are expanded in order of their cumulative log probability. For each node selected for expansion, we generate $N = 64$ candidate tactics from the underlying ReProver model using beam search with default settings, as done in the original ReProver implementation. This forms the ground set for the node, to be sub-sampled by the filtering algorithm. We use beam search decoding because it is deterministic and so ensures that the ground set for a given node remains fixed across runs, allowing us to isolate and compare the effect of the filtering algorithm. The filtering algorithm returns $K$ tactics, which are then executed in the environment and used to update the proof tree, as outlined in A.1. We test three different levels of filtering, with $K \in \{8, 16, 32\}$. Lower values of $K$ correspond to more filtering, for which the choice of filtering algorithm will have a greater impact. We compare the filtering approach of 3D-Prover, as outlined in Algorithm 1, with two baselines. The **Top-K** baseline takes the top $K$ tactics from the ground set as judged by their log probabilities, corresponding to the top $K$ beams. We take $K$ tactics at random from the ground set to form the **Random** baseline, as an exploration-focused comparison.

### 3.2.1 PROOF PERFORMANCE

Table 2 shows the Pass@1 results of our experiments on miniF2F, which is the number of proofs successfully found after a single attempt. We observe that 3D-Prover significantly outperforms both baseline approaches. We also note that Top-$K$ selection performs better than the Random approach, which is unsurprising. The influence of the filtering algorithm becomes more apparent as $K$ is decreased, as there are more tactics filtered out. Our results are consistent with this, with the

| | | miniF2F-test | | | | miniF2F-valid | | |
|---|---|---|---|---|---|---|---|---|
| $K$ | Top-$K$ | Random | 3D-Prover | Gain | Top-$K$ | Random | 3D-Prover | Gain |
| 8 | 22.4 | $19.0 \pm 0.98$ | $\mathbf{24.4 \pm 0.22}$ | +8.9% | 21.7 | 19.3 | **25.0** | +15.2% |
| 16 | 26.5 | $25.4 \pm 0.39$ | $\mathbf{27.3 \pm 0.21}$ | +3.0% | 26.6 | 24.2 | **29.1** | +9.4% |
| 32 | 27.8 | $27.4 \pm 0.26$ | $\mathbf{28.2 \pm 0.25}$ | +1.4% | 27.9 | 27.5 | **28.7** | +2.9% |

Table 2: Percentage of proofs found after one attempt (Pass@1) on miniF2F, with $K$ tactics selected per node, using tactics generated from ReProver. 3D-Prover uses a transition model trained from miniF2F-valid transitions. For miniF2F-test, we report the mean $\pm$ standard error over four runs, with Top-$K$ being deterministic. The Gain column reports the relative improvement over the Top-$K$ baseline. Results for no filtering were 27.8% for miniF2F-test and 27.9% for miniF2F-valid. We observe a clear improvement using 3D-Prover, which increases as more filtering is applied (lower $K$). Our results on miniF2F-test show that 3D-Prover can improve search even for proofs out of distribution of the transition model.

magnitude of improvement given by 3D-Prover increasing for lower values of $K$. 3D-Prover is able to outperform both baselines by providing a tradeoff between the quality, as represented by Top-$K$, and the diversity of the tactics. The choice of $K$ also controls the depth of the proof search, with larger $K$ giving broader search, and smaller $K$ deeper search. As most discovered proofs are short (favouring broad search), the Pass@1 performance for lower values of $K$ is generally lower, however over multiple attempts it can be beneficial to use deeper searches (see Appendix A.2). Finding deep proofs has to date been a significant challenge ( *e.g.* Polu et al. (2022)), with the search tree growing exponentially with the proof depth. The improvement given by 3D-Prover, particularly for deeper search configurations, is a step towards addressing this.

Tree search should be considered as an augmentation of the base model, with the degree of any improvement much smaller than what can be found by improving the generator itself. This is unsurprising, as the generator forms the base set of candidates for the search to explore. Improved search algorithms do however have the advantage of being applicable to different base models, which is important given the rapid advancement of new and better generators. For example, the state-of-the-art DeepSeek-Prover-V1.5 obtains around 2–4% relative improvements in proof success over miniF2F-test with its novel tree search algorithm, compared to no search. In comparison, improving their base model yields a $\sim$36% relative improvement (Figure 5 and Table 1 in (Xin et al., 2024)). Similarly, Table 1 from Polu et al. (2022) shows their search approach yielding 0.04-5.7% relative improvements for miniF2F-valid, with $\sim$40,000 GPU hours required for their best results. We were able to find our improvements with significantly less resources, training our transition model on only a single attempt per proof.

We emphasise that these results were obtained without any hyperparameter tuning, only using the representations as diversity features and model logits as quality scores. We present ablation studies looking closer at these hyperparameters, however a comprehensive sweep is prohibitively expensive with each full attempt taking at least 12h on our hardware. Despite this, we were able to obtain our improvements without any tuning, demonstrating the effectiveness of our approach. Appendix A.3 details the Pass@1 performance over a small set of hyperparameter configurations, where we found no significant improvement over the default $\lambda_s = \lambda_\tau = 0, \theta = 1$. We also highlight that the miniF2F-test results were obtained by training with transitions from miniF2F-valid, showing that 3D-Prover remains effective for proofs out of distribution. The results on miniF2F-valid represent the more common online scenario, with previous attempts on the same dataset being used to improve performance (see, for example, Lample et al. (2022); Polu et al. (2022); Bansal et al. (2019)).

### 3.2.2 ABLATION STUDY

**Effect of the Transition Model** To demonstrate the utility of our transition model representations, we compare to an ablated 3D-Prover where the transition model Encoder is replaced by an Autoencoder of the same size. The Autoencoder is trained to reconstruct the original tactic, and therefore generates representations which reflect only the syntax of the tactic. In this way, we can test our hypothesis that semantically aware tactic representations are useful for proofs, justifying the inclusion

| $K$ | Autoencoder | Transition Model | Gain |
|---|---|---|---|
| 8 | 23.0 | **25.0** | +8.7% |
| 16 | 27.9 | **29.1** | +4.3% |
| 32 | 27.0 | **28.7** | +6.3% |

Table 3: Percentage of proofs found after one attempt (Pass@1) on miniF2F-valid, comparing 3D-Prover with a Transition Model Encoder to an Autoencoder trained to reconstruct the original tactics. We see that 3D-Prover with the Transition Model gives a clear improvement in proof success over the Autoencoder, demonstrating the utility of our representation architecture in Section 2.

of the transition model. As we observe in Table 3, the performance of 3D-Prover with the transition model embeddings is indeed superior to that of the Autoencoder across all values of $K$. This shows that selecting for diversity with respect to the predicted semantics, rather than the syntax, leads to a direct improvement in proof performance.

| | | | 3D-Prover | |
|---|---|---|---|---|
| $K$ | Top-$K$ | Random | $\lambda_s = 0.1$ | $\lambda_s = 0.5$ |
| 8 | $39.0 \pm 0.1$ | $33.4 \pm 0.1$ | $43.3 \pm 0.1$ | $\mathbf{56.5 \pm 0.1}$ |
| 16 | $39.0 \pm 0.1$ | $30.9 \pm 0.1$ | $40.0 \pm 0.1$ | $\mathbf{51.7 \pm 0.1}$ |
| 32 | $35.0 \pm 0.2$ | $29.7 \pm 0.1$ | $35.7 \pm 0.1$ | $\mathbf{41.7 \pm 0.1}$ |

Table 4: Tactic success rate per node for miniF2F-valid (Mean $\pm$ Standard Error), where $\lambda_s$ controls the error weight of quality score in 3D-Prover. No filtering gives $27.7\% \pm 0.2\%$. We see that 3D-Prover leads to fewer errors on average, which can be controlled by increasing $\lambda_s$.

**Success Rate**    We observe from Table 4 that the success rate of tactics chosen by 3D-Prover is significantly improved compared to both baselines. We also note that as $K$ decreases, this improvement increases in magnitude, reflecting the heightened influence of the filtering model. We see that this improvement increases with the error weight $\lambda_s$, which scales the quality scores of tactics by their predicted probability of success. This suggests the error weight term is directly influencing the tactic success rate, showing that it is working as intended.

| | | | 3D-Prover | |
|---|---|---|---|---|
| $K$ | Top-$K$ | Random | $\theta = 1$ | $\theta = 4$ |
| 8 | $83.9 \pm 0.1$ | $88.6 \pm 0.1$ | $90.8 \pm 0.0$ | $\mathbf{91.7 \pm 0.0}$ |
| 16 | $77.5 \pm 0.1$ | $81.4 \pm 0.1$ | $85.9 \pm 0.1$ | $\mathbf{86.6 \pm 0.1}$ |
| 32 | $71.1 \pm 0.1$ | $72.7 \pm 0.1$ | $77.6 \pm 0.1$ | $\mathbf{78.1 \pm 0.1}$ |

Table 5: Percentage of unique environment responses per node in miniF2F-valid (Mean $\pm$ Standard Error). Unique defines either syntactically distinct error messages or responses including at least one previously unseen subgoal. No filtering results in $63.3\% \pm 0.2\%$. $\theta$ controls the temperature of the model scores when calculating quality. We see that 3D-Prover gives a higher diversity of environment responses, increasing with $\theta$.

**Diversity**    To examine diversity, we first look at the percentage of unique environment responses to tactics executed per node, including responses with unique errors (Table 5). As it is difficult to select tactics guaranteed to be successful (Table 4), an exploratory policy should generate tactics which result in more varied outputs, so as to better explore the space. As a second comparison, Table 6 quantifies the likelihood of a successful tactic resulting in a new proof path. We restrict only to successful tactics to account for the discrepancy in success rate between approaches. This gives directly measures the diversity in terms of generating distinct proof paths to explore. In both cases,

| | | | 3D-Prover | |
|---|---|---|---|---|
| $K$ | Top-$K$ | Random | $\theta = 1$ | $\theta = 4$ |
| 8 | $85.3 \pm 0.1$ | $89.9 \pm 0.1$ | $90.1 \pm 0.1$ | $\mathbf{91.1 \pm 0.1}$ |
| 16 | $77.5 \pm 0.1$ | $84.1 \pm 0.1$ | $84.9 \pm 0.1$ | $\mathbf{85.5 \pm 0.1}$ |
| 32 | $72.3 \pm 0.2$ | $76.3 \pm 0.2$ | $76.9 \pm 0.2$ | $\mathbf{77.5 \pm 0.2}$ |

Table 6: Percentage of successful tactics per node resulting in unique subgoal(s) over miniF2F-valid (Mean $\pm$ Standard Error). No filtering gives $67.8\% \pm 0.3\%$. $\theta$ controls the temperature of model scores in 3D-Prover when calculating quality. We observe 3D-Prover results in more unique subgoals per tactic, leading to a more diverse set of proof paths, with larger $\theta$ controlling this.

we see that 3D-Prover results in more diverse responses. As intended, increasing the parameter $\theta$ results in further improvements to diversity under these metrics.

| | | | 3D-Prover | |
|---|---|---|---|---|
| $K$ | Top-$K$ | Random | $\lambda_\tau = 0.1$ | $\lambda_\tau = 1.0$ |
| 8 | $206 \pm 0.8$ | $198 \pm 0.9$ | $155 \pm 0.5$ | $\mathbf{136 \pm 0.5}$ |
| 16 | $220 \pm 0.8$ | $218 \pm 0.9$ | $176 \pm 0.6$ | $\mathbf{152 \pm 0.5}$ |
| 32 | $224 \pm 0.8$ | $215 \pm 0.8$ | $191 \pm 0.7$ | $\mathbf{181 \pm 0.6}$ |

Table 7: Tactic execution time in milliseconds over miniF2F-valid proof attempts (Mean $\pm$ Standard Error). No filtering resulted in $232 \pm 0.9$ milliseconds. $\lambda_\tau$ controls the time weighting of the quality score in 3D-Prover. 3D-Prover selects faster tactics on average, with larger $\lambda_\tau$ magnifying this.

**Execution Time**    Table 7 shows the execution time for tactics over miniF2F-valid transitions. Again we see that 3D-Prover outperforms the baselines, with the improvement increasing with more filtering. Increasing the time weight $\lambda_\tau$ results in further reductions to the average execution time, demonstrating the accuracy of the predictions, and that they can directly result in faster tactics when filtering. In contrast to the diversity or success rate, it is less obvious why we might prefer faster tactics. Lample et al. (2022) (Appendix E) observe that preferring faster tactics can prevent the excessive application of powerful automation tactics such as `simp`. As these generally take longer to run, using faster tactics can help models learn underlying proof arguments which are often hidden by these automations. It can also greatly reduce the number of timeout errors that they cause.

## 4    CONCLUSION

**Future work**    One might consider structured DPPs (Kulesza, 2012), which operate at the tree level to select diverse paths, rather than the node level, which selects diverse edges. Continual learning of the transition model is another avenue, where training on new data as it is generated could lead to more accurate assessments of diversity and quality. Our approach could also be combined with a separate search algorithm such as HTPS (Lample et al., 2022), rather than BestFS. Testing larger models would be a natural extension, for both the transition model and the underlying tactic generator. Our methodology may also be useful to enhance search in domains beyond formal proving, such as code generation or game playing.

**Summary**    We introduce 3D-Prover, a method to augment proof search by filtering candidate tactics to generate diverse and high quality subsets. By generating tactic representations which reflect the response of the proving environment, 3D-Prover is able to filter tactics based on their likely outcome. We evaluate 3D-Prover by augmenting the ReProver LLM on the standard miniF2F benchmark, where we find an improvement in the overall proof success rate (Table 2), particularly for deeper searches. Our ablation studies confirm the utility of our tactic representations, which allow the selection of tactics with improved success rates, diversity, and/or execution time. By effectively pruning the search space, 3D-Prover is a step towards enabling deeper automated proofs.

## 5 REPRODUCIBILITY STATEMENT

We include in our supplementary material all the code necessary to reproduce our results. The README file includes the necessary instructions for setting up the environment, data preprocessing, and the steps required to run our experiments and analysis.

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

## A  APPENDIX

### A.1  PROOF SEARCH SETUP

We first define the space of goals $\mathcal{S}$, tactics $\mathcal{T}$ and failures $\mathcal{F}$. For our purposes, these all contain arbitrary strings, with the goal being a formal proposition, the tactic a command and the failure an error message. We then define the output space as $\mathcal{O} := \mathcal{P}(\mathcal{S}) \cup \mathcal{F}$. A *proof tree* is a DAG $G = (V, E)$ where $V \subset \mathcal{S}$ is the set of goals and $E$ the edges between them. A *proof attempt* for a goal $g_0$ first initialises the proof tree with $V = \{g_0\}, E = \emptyset$. The *search policy* $\pi_S : G \times V \to \mathbb{R}^+$ is a distribution over goals given a proof tree, being used to select a goal $g$ to expand. The *tactic policy* $\pi_T : \mathcal{S} \times \mathcal{T} \to \mathbb{R}^+$ is a distribution over tactics given a goal, where $N \in \mathbb{N}$ tactics are sampled to give tactics $\{t_i\}_{i=1}^N \subset \mathcal{T}$. The goal, tactic pairs $(g, t_i)$ are then passed to the environment $\mathcal{E} : \mathcal{S} \times \mathcal{T} \to \mathcal{O}$. For each pair, after $\tau_i \in \mathbb{R}$ seconds, it returns either a new set of goals $g_i' \subset \mathcal{S}$ or an error, $e_i \in \mathcal{F}$. We define this response as the *output* $o_i \in \mathcal{O}$. We further define the *status* $s_i \in \{0, 1\}$ as 0 if $o_i \in \mathcal{F}$,

1 if $o_i \in \mathcal{P}(\mathcal{S})$ and the *transition* as the tuple $(g, t_i, s_i, \tau_i, o_i)$. The proof tree is then updated with $G = G \cup g_i'$ for all $g_i'$, and the associated transitions are added as edges to $E$. This is repeated until a *proof* is found, or a budget is exhausted. A proof of $g$ is found when $\mathcal{E}(g, t_i) = \emptyset$ for any $t_i$, or if all $\{g_i'\}$ are proven for $\mathcal{E}(g, t_i) = \{g_i'\} \subset \mathcal{S}$. The result of a proof attempt is then the set of transitions $\{(g_k, t_{ki}, s_{ki}, \tau_{ki}, o_{ki})\}$ for all selected goals $g_k$ and their expanded tactics $t_i$. For simplicity, we drop the indices to denote the set of transitions as $\{(g, t, s, \tau, o)\}$.

## A.2 PASS@K

| $K$ | Random | 3D-Prover | Gain |
|-----|--------|-----------|------|
| 8   | 25.7   | **28.6**  | +11.3% |
| 16  | 30.2   | **31.0**  | +2.6% |
| 32  | **29.8** | **29.8** | +0.0% |

Table 8: Percentage of proofs found after four attempts (Pass@4) on miniF2F-test, with $K$ tactics selected per node.

| Pass@$k$ | 3D-Prover | | | Random | | |
|----------|-----------|-----------|-----------|-----------|-----------|-----------|
|          | $K = 8$ | $K = 16$ | $K = 32$ | $K = 8$ | $K = 16$ | $K = 32$ |
| 1 | 24.9 | 27.8 | 28.6 | 18.0 | 21.2 | 28.1 |
| 2 | 26.1 | 29.4 | 29.0 | 22.9 | 28.6 | 29.0 |
| 3 | 26.5 | 29.8 | 29.8 | 24.9 | 29.4 | 29.8 |
| 4 | 28.6 | 31.0 | 29.8 | 25.7 | 30.2 | 29.8 |

Table 9: Pass@$k$ rates for proof attempts on miniF2F-test

Table 8 summarises the Pass@4 results for miniF2F-test, which is the number of proofs found at least once over four attempts, with Table 9 showing the Pass@$k$ up to $k = 4$. We compare 3D-Prover to the Random baseline, taking the same four runs from Table 2, where $\lambda_s = \lambda_\tau = 0$, $\theta = 1$. With Top-$K$ being deterministic, the Pass@$k$ rate is the same as the Pass@1 rate. Given several attempts, $K = 16$ appears to provide a good tradeoff between breadth and depth, performing the best overall. 3D-Prover maintains a large improvement for $K = 8$, with a modest improvement for $K = 16$.

As discussed by Chen et al. (2021), the Pass@$k$ metric favours exploratory approaches as $k$ increases, at the cost of lower performance for smaller $k$. This is because, over many attempts, a highly exploratory approach is more likely to find at least one proof of a given goal, even though it may find fewer proofs in a single attempt than a more exploitative approach. Further discussion in Lample et al. (2022) finds that randomly sampling search parameters also improves Pass@$k$. With Pass@$k$ being expensive to estimate, we fix our parameters over the four runs to give a more accurate estimate of Pass@1. Given this, a large scale experiment sampling these hyperparameters could lead to improved Pass@$k$ results, as Lample et al. (2022) show for their HTPS approach.

## A.3 PROOF SUCCESS RATE OVER HYPERPARAMETERS

Table 10 shows the Pass@1 results on miniF2F-valid for 3D-Prover for our limited hyperparameter sweep. These results suggest that a lower time weight $\lambda_\tau$ leads to better proving results. The diversity parameter $\theta$ hinders performance for the larger value, consistent with what was observed by Chen et al. (2021), where they observe a tradeoff between exploration and Pass@1. Although these parameters may not improve Pass@1, different proofs may favour different configurations, with some requiring *e.g.* more depth or exploration than others. As discussed above, a higher Pass@$k$ can usually be obtained by sampling a wide set of these parameters. For the set of hyperparameters we tested here, we found a cumulative proof rate (or Pass@15) of 32.8% on miniF2F-valid.

|     | $(\lambda_s, \lambda_\tau, \theta)$ | | | | |
| --- | --- | --- | --- | --- | --- |
| $K$ | (0.0, 0.0, 1.0) | (0.1, 0.1, 1.0) | (0.5, 0.1, 1.0) | (0.1, 1.0, 1.0) | (0.1, 0.1, 4.0) |
| 8 | 25.0 | 25.0 | **25.8** | 22.5 | 23.8 |
| 16 | **29.1** | 28.7 | 27.9 | 27.0 | 26.6 |
| 32 | **28.7** | 28.3 | **28.7** | 27.9 | 27.0 |

Table 10: Pass@1 results on miniF2F-valid, over different hyperparameter configurations for 3D-Prover.

## A.4 EMBEDDING DISCUSSION

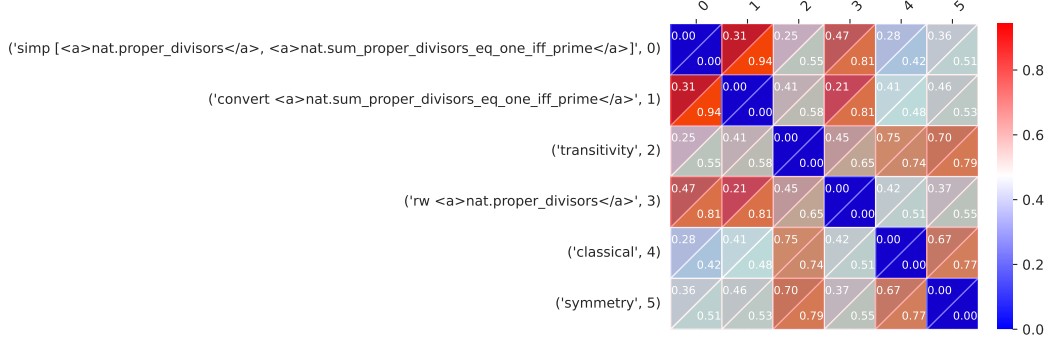

Figure 4: Cosine similarity between tactic embeddings resulting in unique subgoals, for a sample root node in miniF2F-valid. The top value gives the similarity for embeddings from 3D-Prover, while the bottom gives the similarity for embeddings from an Autoencoder. We see that 3D-Prover better separates these semantically distinct tactics, in comparison to the Autoencoder, which only separates based on their syntax.

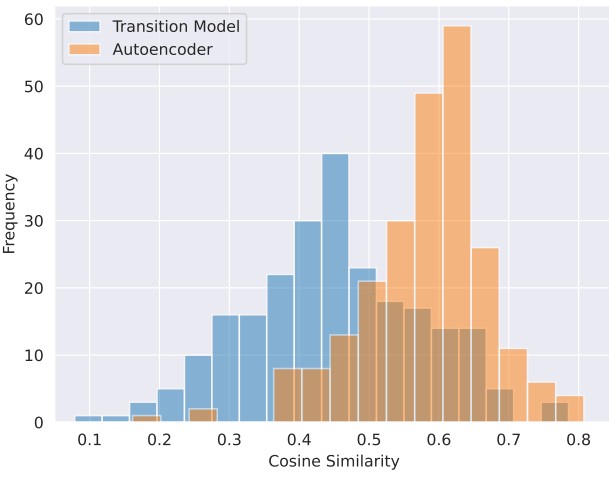

Figure 5: Distribution of cosine similarity for tactic embeddings resulting in unique subgoals, averaged over root nodes in miniF2F-valid. We see that 3D-Prover gives embeddings which better separate these semantically distinct tactics, in comparison the the syntax focused embeddings of the Autoencoder.

**Embedding Comparison** We now investigate whether the transition model (Figure 2) captures tactic semantics rather than syntax in its tactic embeddings. To test this, we examine the cosine

similarity of tactic embeddings which lead to unique subgoals. Figure 4 takes an example node, examining all tactics which lead to a unique subgoal. The upper value displays the cosine similarity given by the transition model, while the lower value displays that given by the Autoencoder in Section 3.2.2. We observe that in most cases, the similarity given by the transition model is much lower than that given by the Autoencoder, which is only considering the syntax of the tactic. For example, the similarity between tactic 3 and 4 is very high for the Autoencoder, given the similar syntax between the two as they use the same lemma. Despite this similar syntax, the transition model embeddings show a high degree of dissimilarity, reflecting the different outcome they have on the environment. We present additional examples in the supplementary code. To generalise beyond these examples, we ran this comparison over the tactic embeddings which lead to unique subgoals for all 244 root nodes in minF2F-valid. Figure 5 shows the distribution of the average cosine similarity for each node, for both the transition model and the Autoencoder. The average cosine similarity for the transition model embeddings was 0.44 while the Autoencoder gave 0.57. While this comparison does not account for similarity between the unique subgoals, it is still clear that the transition model embeddings better separate unique tactics than Autoencoder embeddings which are based on syntax alone. The result of this is a higher likelihood of 3D-Prover selecting tactics which give unique subgoals, which as we show in Section 3.2.2, results in the transition model outperforming the Autoencoder for proof discovery.

**Embedding Objective**   As outlined in Section 2, we train our embeddings to be reflective of the tactic semantics across all three components of Status, Time and Output. Hence 3D-Prover, which selects diverse embeddings, may lead to tactics predicted to have errors, where the errors are diverse in terms of their predicted message. The hyperparameter $\lambda_s$ can alleviate this by weighting the scores based on their likelihood of success. From our experiments (Table 10), there is not necessarily a benefit to Pass@1 by filtering out strongly based on the predicted error likelihood. To speculate, the error prediction, although quite good, is imperfect with many false negatives (Table 1). This can lead to potentially useful tactics being ignored if the error prediction is overly trusted, even though there is a higher tactic success rate overall as in Table 4. Given these prediction errors, it may be the case that selecting goals which are predicted to lead to (diverse) errors may be preferable, given the possibility they result in successful new subgoals. These subgoals may be be quite different from those previously selected, as they are mispredicted, so are clearly outside the space of tactics where the transition model is confident about the outcome. Further analysis could be worthwhile to investigate this. An embedding architecture trained only on successful tactics could be used, however given the high error rate of tactics, this would ignore a large proportion of the transition data.

### A.5   COMPUTATIONAL OVERHEAD

On our hardware, we found 3D-Prover adds a constant overhead, taking approximately 2x as long for tactic generation. The majority of this is in generating embeddings for the 64 tactics, which we were unable to batch on our hardware due to memory constraints. The DPP algorithm itself added almost no overhead once the embeddings were generated. This could be sped up by batching (if memory permits), or through a different architecture. For example, the SEPARATE model in Section 2.2 could be used, where tactics can be batched with much less memory. An augmented architecture which embeds the goal in isolation, which is then given to the tactic encoder as a single vector, could be used. This would provide a speed up while allowing some attention between the tactic and the goal, although not to the degree allowed for by our COMBINED model. As a proof of concept, we used the COMBINED model as it provides the most goal-aware embeddings to test our filtering algorithm.

### A.6   LEANDOJO NOVEL PREMISES EXPERIMENT

We ran an additional experiment on the LeanDojo Novel Premises Yang et al. (2023) benchmark testing 3D-Prover on a larger dataset. This dataset has 2000 evaluation proofs in comparison to the 244 from miniF2F-valid and miniF2F-test, allowing us to evaluate the performance of 3D-Prover on a larger scale.

We trained a transition model from a single ReProver attempt on LeanDojo Novel Premises, before evaluating 3D-Prover with the methodology in Section 3. We set $K$=32 for 3D-Prover, and compare to the model with No Filtering (i.e. $K$=64), and Top-$K$=32. Additionally, we examine the distribu-

tion of proof lengths found from this experiment. To account for different proofs of the same goal, we adjust proof lengths to be the shortest found from any attempt ( e.g. if 3D-Prover finds a proof of length 10, which was found in 3 steps by No Filtering, we count it as length 3). Hence, all proof lengths reported are the shortest found by any method. We report the number of proofs found by each approach, organised by the proof length in Table 11.

| Proof Length | 3D-Prover ($K = 32$) | Top-$K$ ($K = 32$) | No Filtering ($K = 64$) |
|---|---|---|---|
| 1 | 236 | 233 | **237** |
| 2 | 167 | 162 | **174** |
| 3 | **134** | 126 | 131 |
| 4 | **60** | **60** | 54 |
| 5 | **40** | 29 | 24 |
| 6 | **7** | 6 | 2 |
| 7 | **2** | 0 | 0 |
| Total | **646** | 626 | 622 |
| Pass@1 | **32.3%** | 31.3% | 31.1% |

Table 11: Number of Proofs found on LeanDojo Novel Premises, sorted by proof length.

obtains a relative improvement of 3.2% over Top-$K$, and a 3.9% relative improvement over No Filtering in terms of the number of proofs found. We see that 3D-Prover finds deeper proofs, while maintaining a high proof success rate for shallower proofs, unlike Top-$K$. The No Filtering approach, as expected, finds the most shallow proofs, however quickly drops off in performance for deeper proofs. We also note that 3D-Prover found the 2 longest proofs of length 7, with neither baseline finding any.

