# OpenReview forum: "3D-Prover: Diversity Driven Theorem Proving With Determinantal Point Processes"
_ICLR.cc/2025/Conference — Submitted to ICLR 2025_

### Official Review · Reviewer_xeEr · 2024-10-30

**Soundness:** 4
**Presentation:** 3
**Contribution:** 3
**Rating:** 8
**Confidence:** 4

**Summary:**

The paper introduces 3D Prover, a framework that combines a representation learning algorithm to generate semantically aware encodings of proof states and tactics with a filtering algorithm for enhancing semantic diversity in LLM-generated tactics for proof synthesis. The proposed method was tested on top of ReProver, demonstrating its effectiveness. Additionally, ablation studies validate the contribution of each component in the design.

**Strengths:**

- The proposed algorithm, 3D Prover, is well-motivated and practical. The paper presents an approach to augmenting proof search history using an existing LLM to enhance tactic selection, resulting in more efficient navigation within the search space.
- The semantic-aware representation learning of proof states and tactics is well-established and analyzed. The representations learned from proof history are useful for downstream tasks, including proof synthesis. The experimental design and results demonstrate the impact of each component of the proposed learning framework.
- The proposed filtering framework is straightforward and effective, with results outperforming ReProver without modifying the underlying model. This approach is thus beneficial for advancing proof synthesis in large language models.

**Weaknesses:**

- Section 3.1 could benefit from greater clarity. Specifically, the motivation behind using this method could be expanded upon - why was DPP chosen over other sampling algorithms for balancing quality and exploration in this context?
- In the experiments, both $\lambda_s$ and $\lambda_{\tau}$ are set to zero. Additionally, Section A.2’s hyperparameter sweep further suggests that these terms (or at least $\tau_i$) do not significantly contribute to filtering performance. Although this result is not unexpected, it should be explicitly discussed in the experiments section to enhance the clarity of the findings.

**Questions:**

I wonder what the intuition is for preferring shorter execution time for tactic. Tactic execution time can vary based on factors like the goal state; for example, `linarith` may require different durations depending on the complexity of the arithmetic expressions involved. Moreover, while tactics like `linarith` can take longer for more complex arithmetic, they are often the very effective in reducing the theorem down to its underlying logic.

---

> ### Author Response · Authors · 2024-11-19
>
> Thank you for the thorough and helpful review. We are glad that you found our approach well motivated and practical, and
> useful for advancing proof synthesis in large language models. We hope to address your concerns and questions in order
> below.
>
> # Q.1
>
> *Section 3.1 could benefit from greater clarity. Specifically, the motivation behind using this method could be expanded
> upon - why was DPP chosen over other sampling algorithms for balancing quality and exploration in this context?*
>
> We chose DPP due to the inherent trade-off between diversity and quality, and its simplicity.
> We are unaware of other sampling algorithms which
> do this inherently (although we appreciate any alternative suggestions!).
> Given DPPs are used in a variety of areas to successfully increase diversity, such as
> document summarization, pose estimation and video summarisation, they are a natural choice [1].
> We agree this context would help clarify the motivation behind DPP, so we will include this in the revision.
>
> References:
>
> [1] Elfeki et al., GDPP: Learning Diverse Generations using Determinantal Point Processes, https://arxiv.org/pdf/1812.00068
>
> # Q.2
>
> *In the experiments, both $\lambda_s$ and $\lambda_\tau$ are set to zero. Additionally, Section A.2’s hyperparameter sweep further suggests that
> these terms (or at least $\tau_i$) do not significantly contribute to filtering performance. Although this result is not
> unexpected, it should be explicitly discussed in the experiments section to enhance the clarity of the findings.*
>
> We will include a brief discussion of the minimal impact of hyperparameters to the overall proving performance in the
> experiments section, to enhance the clarity.
>
> # Q.3
>
> *I wonder what the intuition is for preferring shorter execution time for tactic. Tactic execution time can vary based
> on factors like the goal state; for example, `linarith` may require different durations depending on the complexity of the
> arithmetic expressions involved. Moreover, while tactics like `linarith` can take longer for more complex arithmetic, they
> are often the very effective in reducing the theorem down to its underlying logic.*
>
> There is a good discussion on this point in Appendix E from [1], where it is observed that optimising for faster tactics
> leads to improved performance.
> Firstly, they find that models can become over reliant on automation tactics such as `ring`, `linarith` and `simp`, which are powerful but
> can prevent the model from learning the underlying arguments of the proofs.
> As these generally take much longer to execute than tactics such as rewrite or apply,
> they show that minimising the execution time is a way to prevent over reliance on these tactics.
> Additionally, these tactics often result in timeout errors, wasting valuable search resources. By selecting tactics which
> generally run faster, the number of timeout errors is greatly reduced.
>
> We will add this discussion to the revision, to provide more context for the choice of execution time as a filtering criterion.
>
> References:
>
> [1] Lample et al., HyperTree Proof Search for Neural Theorem Proving, https://arxiv.org/pdf/2205.11491

---

> > ### Comment · Reviewer_xeEr · 2024-11-19
> >
> > I appreciate the authors' comprehensive response to my concerns. They have addressed all points raised in my review, particularly through their explanation of DPP's motivation with relevant references, and their discussion on tactic execution time effects. Given these thorough and satisfactory responses, along with the authors' commitment to incorporating these clarifications in their revision, I have increased my score to 8 (accept).

---

### Official Review · Reviewer_hTv9 · 2024-11-03

**Soundness:** 2
**Presentation:** 3
**Contribution:** 3
**Rating:** 5
**Confidence:** 4

**Summary:**

- This paper proposes a method to filter tactics generated by sampling a tree search policy based on diversity in formal theorem proving. To capture tactic diversity, it uses an encoder to embed both the tactic and the goal it is applied to, followed by Determinantal Point Processes (DPP) to select tactics based on these embeddings.
- The encoder used for embedding is trained on two tasks: (a) an auxiliary decoder that applies cross-attention on the embedding to predict the environment’s response after applying the tactic (yielding the next goal if successful, or error messages if unsuccessful); (b) a single-layer MLP that predicts both the success of the tactic and the time required by the Lean prover to verify it. In the final tactic selection step, embeddings are also weighted according to predictions from the MLP regarding tactic success and Lean prover verification time.
- Experimental results show that the tactics selected by 3D-Prover achieve a significantly higher pass rate compared to the baseline.

**Strengths:**

1. This paper introduces an additional filtering mechanism within language model-based tactic tree search, providing a fresh perspective for research on tree search in theorem proving.
2. Experimental results demonstrate noticeable performance improvements even with a relatively small number of filtered tactics, suggesting that the diversity-based screening method is effective for scenarios with limited computational resources.

**Weaknesses:**

1. The paper filters tactics only after they have been sampled by a language model, so the computational cost of model sampling remains unchanged. While the paper tries to reduce Lean prover calls with a specialized filtering algorithm, it does not provide a comparative analysis on this aspect. Does 3D-Prover achieve better results than the case that these computational resources were used to apply the Lean prover to all tactics the language model generated?

2. The paper emphasizes balancing quantity and diversity in tactic selection from the sampled tactics. However, given limitations in the language model’s capability, it may not generate tactics diverse enough to meet the problem’s requirements. This suggests that the proposed method may not fundamentally improve the success rate beyond the language model’s own capacity, limiting 3D-Prover’s primary advantage to efficiency (though this, as noted in *Weakness 1*, requires further estimation).

**Questions:**

Please refer to the first point in Weaknesses.

---

> ### Author Response · Authors · 2024-11-19
>
> Thank you for the review and constructive feedback.
> We appreciate your comments that our approach provides
> a fresh perspective on tree search in theorem proving.
> We hope to address your concerns regarding the soundness of our approach, specifically
> regarding comparative computational cost and the limitations of the underlying language model.
> These are valid and reasonable concerns, however we would like to highlight that both of these limitations are inherent
> to much of the research in this area, as we will detail below.
>
> ## Q.1
>
> *The paper filters tactics only after they have been sampled by a language model, so the computational cost of model
> sampling remains unchanged. While the paper tries to reduce Lean prover calls with a specialized filtering algorithm, it
> does not provide a comparative analysis on this aspect. Does 3D-Prover achieve better results than the case that these
> computational resources were used to apply the Lean prover to all tactics the language model generated?*
>
> A comparison reallocating resources in this way would be quite difficult, given that the provers and the tactic generator/filters
> are implemented as separate processes.
> Similar to [1], we have a set of tactic generators serving requests from a separate set of proving processes
> (2 generators/filters serving 4 proving processes in our case).
>
> To control for the underlying hardware setup, it is standard practice in the area to compare approaches based on the
> number of proofs achieved with a fixed environment budget (e.g. [1, 2, 3, 4, 5]).
> Additional search time is not considered or reported in any of these approaches,
> which all perform additional computations beyond the underlying tactic generator.
> As we discuss from line 408, our search method gives a similar magnitude of improvement to approaches
> which use significantly more resources in their search algorithms.
> Our embedding model is also significantly cheaper than an LLM call (as used for search in e.g. [3, 5]),
> and we discuss this overhead in Appendix A.5.
>
> Given the exponential growth of the search tree, the primary benefit of our approach is to enable deeper proofs
> which are not feasible (under the same environment budget) for broader searches without filtering, while
> maintaining a high proof success overall. Our additional results in G.2 support this further, where we can see that
> applying all tactics the language model generated (No Filtering) leads to significantly fewer proofs greater than length 3.
> To reach proofs of length 6 or 7, as achieved by 3D-Prover, would require a significantly larger environment budget
> if all tactics were applied, considering the exponential growth of the search tree.
>
> References:
>
> [1] Bansal et al., HOList: An Environment for Machine Learning of Higher Order Theorem Proving, https://arxiv.org/pdf/1904.03241
>
> [2] Xin et al., DeepSeek-Prover-V1.5: Harnessing Proof Assistant Feedback for Reinforcement Learning and Monte-Carlo Tree Search, https://arxiv.org/pdf/2408.08152
>
> [3] Polu et al., Formal Mathematics Statement Curriculum Learning, https://openreview.net/pdf?id=-P7G-8dmSh4
>
> [4] Wu et al., TacticZero: Learning to Prove Theorems from Scratch with Deep Reinforcement Learning, https://arxiv.org/pdf/2102.09756
>
> [5] Lample et al., HyperTree Proof Search for Neural Theorem Proving, https://arxiv.org/pdf/2205.11491
>
> ## Q.2
>
> *The paper emphasizes balancing quantity and diversity in tactic selection from the sampled tactics. However, given
> limitations in the language model’s capability, it may not generate tactics diverse enough to meet the problem’s
> requirements. This suggests that the proposed method may not fundamentally improve the success rate beyond the language
> model’s own capacity, limiting 3D-Prover’s primary advantage to efficiency (though this, as noted in Weakness 1,
> requires further estimation).*
>
> You are correct to point out that proof success will be limited by the underlying language model.
> As we discuss from line 408, this is a limitation of any search approach,
> and would apply to all other search algorithms in the domain.
> Improved search approaches do however have the advantage of being independent of the tactic generator,
> so they can be used with newer models as they improve. As we mention above, the primary advantage of 3D-Prover is
> enabling new and deeper proofs to be discovered.

---

> > ### Comment · Reviewer_hTv9 · 2024-11-27
> > **Thank you for your response.**
> >
> > Thank you for your response. I still firmly believe that quantitative comparative experiments are necessary. As you also agreed in your response to *Weakness 2*, this paper fundamentally does not improve the *effectiveness* of language model-generated correct theorems but can at most enhance *efficiency* through pruning in tree search. The reason prior works did not provide search time comparisons is that this was not their core contribution. I believe this paper would still benefit from a quantitative comparative experiment. As I described in my response to the Associate Program Chairs, this experiment is very simple: it only requires comparing the execution time efficiency of the method proposed in this paper versus baseline methods. This result determines whether the improvements achieved by this paper stem from additional computations that were performed but not fairly compared in terms of computation budget.

---

> ### Author Response · Authors · 2024-12-02
>
> Thank you for the follow up response, and your active engagement in the discussion.
> We are unsure of what you mean by **effectiveness** vs **efficiency**, we focus on the standard metrics
> of proof success (as defined by Pass@1), for which we show an improvement.
> Interpreting efficiency as the number of proofs found per wall clock time,
> your proposed experiment could provide an additional perspective, and we give additional results below.
> However, we disagree that the absence of such a comparison is a major weakness of the paper.
> We again emphasise that our approach was compared
> against baselines using the same methodology (i.e. using a fixed environment budget)
> as related work, which we detailed in our original response.
>
> We firmly disagree on the point that **The reason prior works did not provide search time comparisons is that this
> was not their core contribution.**
> For example, the primary algorithmic innovation of Lample et al. [1] is their development of the MCTS inspired
> HTPS algorithm, which is a novel search approach, as is our method.
> Similarly, Polu et al. [2] and Wu et al. [3] both purport improvements from
> novel search approaches, which they evaluate in comparison to baselines without controlling for additional search
> computations.
> The fact that these papers had additional contributions does not impact the soundness of the evaluation of their search
> approaches.
>
> As we have a set of GPU tactic generators + filters serving requests from a separate set of CPU proving clients, which
> then
> execute the tactics in Lean, we cannot easily reallocate resources from one to the other.
> In Appendix A.5, we discuss the overhead of our filtering approach, which is less than the cost of an LLM call as is used by other search approaches (e.g. [1, 2]), but varies significantly based on the underlying
> hardware setup.
> For example, given additional GPU memory, we could significantly speed up filtering by batching tactic
> embeddings.
> This GPU memory would be of no benefit to a model without filtering, as it could not be used to speed up the tactic
> generation or the proving clients.
> Combined with the asynchronous setup of the provers and tactic generators, this makes comparisons based on execution wall time highly hardware dependent.
>
> We do however acknowledge your strong preference for such a comparative experiment.
> Given this, we ran a single pass over miniF2F-valid for the Top-K baseline and 3D-Prover,
> where we include the tactic generation and filtering time (in the case of 3D-Prover) in the 600 second budget, which
> took approximately 3 days to finish on our hardware.
> The number of proofs found from each approach, the Pass@1 percentage, and relative improvement is given in the table below:
>
> | K    | Top-K      | 3D-Prover      | Gain  |
> |------|------------|----------------|-------|
> | K=8  | 52 (21.3%) | **58 (23.8%)** | 11.5% |
> | K=16 | 63 (25.8%) | **65 (26.6%)** | 3.2%  |
> | K=32 | 66 (27.0%) | **67 (27.5%)** | 1.5%  |
>
> We still see an improvement given by 3D-Prover, particularly for deeper searches.
> The magnitude of the improvement is decreased, however as discussed in A.5, the majority of the
> filtering time is used for tactic embeddings, so we expect these 3D-Prover results would
> improve further with batching if given more GPU memory (which would not help Top-K). As we also discuss in A.5,
> there could be further speed improvements with different transition model architectures.
> As a proof of concept, we used the most performant transition model architecture from Section 2 for 3D-Prover, evaluating it with respect to environment budget, in line with related work.
> We are however happy to include some of this discussion, as well as the results above, as an extension to A.5 if you
> believe it will benefit the paper.
>
> References:
>
> [1] Lample et al., HyperTree Proof Search for Neural Theorem Proving, https://arxiv.org/pdf/2205.11491
>
> [2] Polu et al., Formal Mathematics Statement Curriculum Learning, https://openreview.net/pdf?id=-P7G-8dmSh4
>
> [3] Wu et al., TacticZero: Learning to Prove Theorems from Scratch with Deep Reinforcement
> Learning, https://arxiv.org/pdf/2102.09756

---

### Official Review · Reviewer_1vwm · 2024-11-05

**Soundness:** 3
**Presentation:** 3
**Contribution:** 2
**Rating:** 5
**Confidence:** 4

**Summary:**

The paper presents a method to address a key issue in automated theorem proving: the vast search space that grows exponentially with proof depth due to numerous potential tactics. To manage this complexity, the authors introduce 3D-Prover, a filtering mechanism for proof tactics that prioritizes diversity and quality using Determinantal Point Processes (DPPs). The method involves two parts:
  - Tactic Representation Learning: The authors generate semantic representations of tactics to predict their impact on the proving environment, likelihood of success, and execution time. This prediction model uses past proof attempts (synthetic data) to form representations of tactics based on their effect rather than mere syntactic similarity.
  - Filtering Mechanism: Using DPPs, 3D-Prover filters the tactic pool, selecting those that are both high quality and semantically diverse. This selection optimizes proof success while minimizing redundant tactic exploration.

This paper augments the ReProver LLM (proposed in an earlier paper) by introducing a tactic filtering mechanism (3D-Prover) at each step of the proof search, whereby the tactic space is reduced. The authors use Best-first Search for proof search where nodes are expanded in order of their cumulative log probability. Specifically, for each tactic state, the ReProver LLM is used to sample 64 candidate tactics. From the 64 tactics, the proposed tactic-filtering mechanism is used to select a subset of K (set at 8, 16 or 32) tactics. A lower value of K indicates a strong filtering and is aimed to cut down the proof search even more. The authors compared their filtering (from 64 tactics to K = 8, 16 or 32 tactics) with two baselines viz., Top-K (selects top K tactics from 64 candidate tactics, based on their log probabilities) and Random (selects K tactics at random from the 64 candidate tactics).

3D-Prover is tested on the miniF2F-test and miniF2F-valid benchmark, where the basic prover framework is the same as the ReProver LLM, and 3D-Prover is used as a tactic filtering step at each state. The authors claim that filtering tactics results in an increased proof success rates due to reduced execution time, and more diverse tactic sets. The approach demonstrates scalability and effectiveness for deeper, complex proofs, addressing challenges faced by other models like DeepSeek-Prover-V1.5 and contributing to proof success beyond the generator's baseline. The experiments indicate that 3D-Prover outperforms traditional Top-K or Random tactic selection, when 3D-Prover filtering is applied.

Notably, from Table 2, it is evident that for the Top-K baseline, the pass@1 metric i.e., the number of proofs found by ReProver after one attempt is 22.4%. And by incorporating filtering by 3D-Prover with K=8 (i.e., a stricter tactic filtering), pass@1 increases to 24.4%. However for K=32, the pass@1 metric increases minutely from 27.8% to 28.2% because of tactic filtering. A Random filtering, as expected, performs worse than Top-K baseline or the proposed 3D-Prover filtering. The authors also perform ablation study to demonstrate that the filtered tactics are indeed diverse.

The authors claim that 3D-Prover's design, which layers a diversity-driven tactic filter atop conventional LLMs, points to a promising direction for automated theorem proving, enabling faster, deeper proofs while controlling for computational resources.

**Strengths:**

- The paper introduces a novel approach to tactic filtering that emphasizes both diversity and quality. The concepts of transition-aware tactic embeddings and the use of Determinantal Point Processes (DPPs) are innovative in the context of theorem proving, although DPPs themselves are a well-established probabilistic method and not a new contribution of this paper.

- With diversity-driven tactic filtering, the Best-First Search can explore deeper proofs, leading to a faster proof search overall. This is impressive. However, I did not find a comparison of execution times with and without filtering. Table 7 compares execution times with other baselines, but the paper does not clarify the improvement in proof search time specifically due to filtering.

- The tactic filtering method discards a significant portion (87.5%) of sampled tactics from the ReProver LLM for each tactic state, with the proposed 3D-Prover filtering approach outperforming baseline methods like Top-K and Random filtering. By effectively reducing the tactic space, this filtering approach enables more efficient proof searches with fewer resources. While this is a strong result, it would be even more compelling if the paper included a comparison against a no-filtering setup (i.e., considering all 64 tactics).

**Weaknesses:**

- The preliminaries and transition models presented in Section 1.2 and 2.1 could have been written in a simplified manner. Some notations are hard to follow and expressed in a circuitous way.

- Why is the decoder taking as input a concatenation of e and g, when e is already an encoded and pooled version of t and g? What is the motive of having g twice?
Also, e is an encoded version. How is concatenating an encoded string to a non-encoded string g, meaningful?

- The authors use metrics like BLEU, ROUGE-L-F1 and top-4 accuracy (proportion of samples which have one beam identical to the ground truth) to evaluate the "Output" in Figure 2. BLEU and ROUGE are similarity metrics. How is checking similarity with the ground-truth "Output" relevant? Because even if the generated "Output" and the ground-truth "Output" are very close according to these similarity scores, they may not be a syntactically and semantically correct next tactic state.

- In lines 408-418, I do not see the quoted values in the referred tables: "∼36% relative improvement (Table 1)" and "∼6–9% relative improvements (Table 7)". It is not clear if these refer to the tables in the respective papers.

- I have concerns about the training dataset used for the Transition Model. Specifically, in lines 220-222, it is mentioned that "We obtain the dataset D from a vanilla ReProver attempt on miniF2F-valid, which results in 498,236 transitions, which we split randomly into 95% training, 5% testing." i.e., 3D-Prover uses a transition model trained from miniF2F-valid transitions. In that case, doesn't that mean that in Table 2, the proof search results for miniF2F-valid, are trained and evaluated on the same dataset? And for this reason, the results for miniF2F-valid in Table 2 are significantly better than those for miniF2F-test?
Further, the improvement in pass@1 values on the miniF2F-test is not that significant. For K=8, there is a 22.4% to 24.4% i.e. a 2% increase in the number of proofs found. And this improvement goes down to 0.4% (27.8% to 28.2%) for K=32.
Also, the paper does not provide the results without any kind of filtering.

- One of the most concerning weaknesses of this paper is that it does not compare the ReProver (for tactic prediction) + 3D-Prover (for tactic filtering) framework against any state-of-the-art method like DeepSeek-Prover, InternLM-Math, etc. The filtering method is only compared with two baseline tactic filtering mechanisms i.e., Random filtering and top-k filtering. But the paper does not show whether the effect of this filtering is significant enough, such that it outperforms other state-of-the-arts that do not use filtering or use some kind of state-of-the-art filtering.

Although the 3D-Prover introduces a novel design by layering a diversity-driven tactic filter on top of conventional LLMs, the paper falls short in evaluating and demonstrating its superiority over state-of-the-art theorem provers. Additionally, the results show only minimal improvement compared to baseline tactic filtering methods, such as Top-K and Random filtering.

**Questions:**

Please address the issues raised in the Weaknesses section.

---

> ### Author Response · Authors · 2024-11-19
>
> We thank you for the very detailed and constructive review.
> We appreciate your comment that our approach enabling deeper proofs is impressive,
> and that we have a strong result in enabling more efficient proof search.
> We address your concerns in order below.
>
> # Q.1
>
> *With diversity-driven tactic filtering, the Best-First Search can explore deeper proofs, leading to a faster proof
> search overall. This is impressive. However, I did not find a comparison of execution times with and without filtering.
> Table 7 compares execution times with other baselines, but the paper does not clarify the improvement in proof search
> time specifically due to filtering.*
>
> To clarify, the benefit of our approach is not necessarily to make proof search faster. It is to facilitate the
> discovery of new proofs by improving the search algorithm of a base model, given the same environment budget. If
> desired, our approach can however optimise for execution time as seen in Table 7.
>
> We agree that we should include a comparison of execution times to the no filtering setup, which we will add to the
> revision. The average execution time for tactics without filtering was 232 milliseconds (plus/minus 0.9), which was longer
> than any of the filtering methods. We will also include the no filtering numbers for the other ablations (i.e. Table 4,5,6),
> as we agree that this is a useful comparison to make.
>
> # Q.2
>
> *While this is a strong result, it would be even more compelling if the paper included a comparison against a
> no-filtering setup (i.e., considering all 64 tactics).*
>
> Thank you for this suggestion, as it is a reasonable and important comparison to include, which will make our
> results more compelling.
> We will include this in the revised version, but to summarise here, the pass@1 results for no filtering (i.e. top-K=64) are 27.8\% for miniF2F-test and
> 27.9\% for miniF2F-valid. These happen to be the same results as for Top-K=32, for which we show an improvement in
> performance as seen in Table 2. As we mention above, we will also include the no-filtering results for our ablation studies.
>
> We also show in G.2, for a larger dataset (LeanDojo Novel Premises), a 3.9\% relative improvement over no filtering,
> with a particularly large increase in the number of deep proofs discovered. This further supports our approach
> when compared to no filtering.
>
>
> # Q.3
> *The preliminaries and transition models presented in Section 1.2 and 2.1 could have been written in a simplified manner.
> Some notations are hard to follow and expressed in a circuitous way.*
>
> We can see how some of the notation might be difficult to follow, as we have tried to be as precise as possible.
> We believe that moving some of the more detailed notation to an appendix would help simplify this, which we will do for
> the revision.
>
> # Q.4
>
> *Why is the decoder taking as input a concatenation of e and g, when e is already an encoded and pooled version of t and
> g? What is the motive of having g twice?*
>
> As with reviewer HiN1, we agree that some additional motivation and explanation of the transition model architecture
> would be beneficial, which we cover in G.1. We expand on this here to address your specific question.
>
> To ensure that we generate a useful tactic representation in e, the only tactic information we allow
> the Decoder is through e.
> The goal g is included in e as it allows for improved, goal-aware tactic representations.
> This approach gives tactic representations which can reflect the context they are applied in.
> For example, if the goal state has a lemma which is referenced by the tactic, then the encoder will benefit by having
> access to the goal when generating the encoding.
> As we show in Table 1, if we don't include the goal in the tactic encoding, then the performance is greatly reduced (
> COMBINED vs SEPARATE).
>
> As we only pool the tactic tokens (after they have attended to the goal) to generate
> e (Figure 2),
> there is a large loss of information about the goal in the encoding.
> We therefore also provide the Decoder with the original goal tokens, so that it has full access to the original goal for the output
> prediction. We can run an ablation removing this, if you think it will be useful, although the NO TACTIC baseline in
> Table 1 shows the performance if we give the Decoder g alone, without the (goal, tactic) representation.
>
>
> *How is concatenating an encoded string to a non-encoded string g, meaningful?*:
>
> The input to the Decoder will be the tactic encoding vector, concatenated with the token vectors for the original goal
> from the embedding matrix of the Decoder. This is meaningful as the Decoder can now use information from the tactic (through
> the encoding e) as it attends to the tokens of the original goal, allowing it to better predict the output than if it has
> no tactic information (as we show in the NO TACTIC baseline in Table 1).

---

> ### Author Response · Authors · 2024-11-19
>
> # Q.5
>
> *The authors use metrics like BLEU, ROUGE-L-F1 and top-4 accuracy (proportion of samples which have one beam identical
> to the ground truth) to evaluate the "Output" in Figure 2. BLEU and ROUGE are similarity metrics. How is checking
> similarity with the ground-truth "Output" relevant? Because even if the generated "Output" and the ground-truth "Output"
> are very close according to these similarity scores, they may not be a syntactically and semantically correct next
> tactic state.*
>
> We agree that a drawback of these metrics, as you point out, is that they only compare the lexical similarity of two
> sequences.
> There may be similar sequences which have quite different semantics, or are not valid tactic states.
> Regardless, they are still useful (although imperfect) metrics, as they demonstrate how well our model
> can predict the environment output without actually executing the tactic.
> Beyond these metrics, we also demonstrate that our tactic representations effectively capture the semantics of tactics
> in Appendix A.3.
>
> We also note that these metrics are used in other domains where semantic similarity is desired, such as translation and
> code completion. For example, Figure 9 in the survey from [1] finds that BLEU, ROUGE and Top-k are among the most widely
> used metrics for evaluating code completion.
>
> As far as we are aware, there is no simple evaluation metric for capturing semantic similarity (we are open to test
> other metrics, if you have suggestions).
> Given this, we did initially investigate an "Autorater" approach (discussed in [2]),
> where we have a high performance LLM (Gemini-Pro) evaluate predictions in terms of their semantics.
> We didn't include these results in the paper as we thought it would distract from the main results, as it is somewhat
> lengthy to cover.
> To help address your concern, we present an overview of the approach and results here, which we can include
> as an appendix in the revision if you find it useful:
>
> We evaluated the Output predictions for the COMBINED vs SEPARATE models in Section 2 by asking the Gemini-Pro LLM to
> score both in terms of their semantics. The prompt used, along with an example being assessed, is given to LLM as below:

---

> ### Author Response · Authors · 2024-11-19
> **Q.5 Autorater example**
>
> ```
> You are an expert in Lean 3 theorem proving. Your task is to evaluate and rank the quality of two predictions, where each prediction is the result of applying a given tactic to a given goal. Your evaluation should be based on how close the prediction is to the true outcome, semantically. A closer prediction syntactically is not necessarily better. For example, if the prediction is a negation of the true outcome, it should be ranked lower than a prediction that is a conjunction of the true outcome. The input first contains the original goal, which is a list of premises, followed by the goal(s) to prove (these are lists of hypotheses, followed by the "⊢" character and the goal itself). Then you are given the applied tactic, the true outcome, and two predictions. The output should be your reasoning, followed by a ranking of the two predictions, giving a score from 0 to 5 to each.
>
> Your reasoning should first explain the goal, the tactic, and the true outcome. Then, you should explain the two predictions, and how they relate to the true outcome. Finally, return your output as a list of two scores, where the first score is for the first prediction, and the second score is for the second prediction.
>
> Goal:
>
> case list.cons
> α : Type u_1
> i : ℕ,
> j : α,
> l : list α,
> h : i < (j :: l).to_buffer.size,
> `h' : i < (j :: l).length`*
> ⊢ (j :: l).to_buffer.read ⟨i, h⟩ = (j :: l).nth_le i h'
>
> Tactic: `induction l with hd tl IH`
>
> True Outcome:
>
> case list.cons, list.nil
> α : Type u_1,
> i : ℕ,
> j : α,
> h : i < [j].to_buffer.size,
> h' : i < [j].length
> ⊢ [j].to_buffer.read ⟨i, h⟩ = [j].nth_le i h'
>
> case list.cons, list.cons
> α : Type u_1,
> i : ℕ,
> j hd : α,
> tl : list α,
> IH : ∀ (h : i < (j :: tl).to_buffer.size) (h' : i < (j :: tl).length), (j :: tl).to_buffer.read ⟨i, h⟩ = (j :: tl).nth_le i h',
> h : i < (j :: hd :: tl).to_buffer.size,
> h' : i < (j :: hd :: tl).length
> ⊢ (j :: hd :: tl).to_buffer.read ⟨i, h⟩ = (j :: hd :: tl).nth_le i h'
>
> Prediction 1:
> case list.cons, list.nil
> α : Type u_1,
> i : ℕ,
> j : α,
> h : i < [j].to_buffer.size,
> h' : i < [j].length
> ⊢ [j].to_buffer.read ⟨i, h⟩ = [j].nth_le i h'
>
> case list.cons, list.cons
> α : Type u_1,
> i : ℕ,
> j hd : α,
> tl : list α,
> IH : ∀ (h : i < (j :: tl).to_buffer.size), i < (j :: tl).length → (j :: tl).to_buffer.read ⟨i, h⟩ = (j :: tl).nth_le i h',
> h : i < (j :: hd :: tl).to_buffer.size,
> h' : i < (j :: hd :: tl).length
> ⊢ (j :: hd :: tl).to_buffer.read ⟨i, h⟩ = (j :: hd :: tl).nth_le i h'
>
> Prediction 2:
>
> case list.cons, list.nil
> α : Type u_1,
> i : ℕ,
> j : α,
> h' : i < [j].length,
> h : i < [j].to_buffer.size
> ⊢ [j].to_buffer.read ⟨i, h⟩ = [j].nth_le i h'
>
> case list.cons, list.cons
> α : Type u_1,
> i : ℕ,
> j hd : α,
> tl : list α,
> IH : ∀ (h : i < (j :: tl).to_buffer.size), (j :: tl).to_buffer.read ⟨i, h⟩ = (j :: tl).nth_le i h',
> h : i < (j :: hd :: tl).to_buffer.size,
> h' : i < (j :: hd :: tl).length
> ⊢ (j :: hd :: tl).to_buffer.read ⟨i, h⟩ = (j :: hd :: tl).nth_le i h'
> ```
>
> The LLM response for this was:
>
> ```
> The goal is to prove that `(j :: l).to_buffer.read ⟨i, h⟩ = (j :: l).nth_le i h'` where `j : α`, `l : list α`, `i : ℕ`, `h : i < (j :: l).to_buffer.size`, and `h' : i < (j :: l).length`. In other words, we want to show that reading the `i`-th element of the buffer obtained by converting the list `j :: l` is the same as getting the `i`-th element of the list directly using `nth_le`.
>
> The tactic `induction l with hd tl IH` applies induction on the list `l`. This will generate two goals:
>
> 1. **Base Case (list.nil):** Prove the goal for `l = []`.
> 2. **Inductive Step (list.cons):** Assuming the goal holds for `l = tl`, prove it for `l = hd :: tl`.
>
> The true outcome is the expected result of applying the `induction` tactic, generating the base case and inductive step with the correct assumptions and goals.
>
> Let's analyze the predictions:
>
> *Prediction 1:*
>
> **Base Case:**  It's identical to the true outcome.
> **Inductive Step:** It makes a subtle error in the inductive hypothesis (IH). The true outcome has `∀ (h : i < (j :: tl).to_buffer.size) (h' : i < (j :: tl).length)`, which quantifies both `h` and `h'` universally. Prediction 1 uses `∀ (h : i < (j :: tl).to_buffer.size), i < (j :: tl).length →`, making the dependency between `h'` and the conclusion implicit instead of explicit.
>
> *Prediction 2:*
>
> **Base Case:** It swaps the positions of `h` and `h'` in the context, which is semantically irrelevant and doesn't affect the proof.
> **Inductive Step:** It makes a more serious error by completely omitting `h'` from the inductive hypothesis. This leads to a weaker IH that cannot be used to prove the inductive step.
>
> **Ranking:**
>
> **Prediction 1:** 4/5 - It gets the base case right and the inductive step almost correct. The error in the IH is subtle and could potentially be worked around.
> **Prediction 2:**  2/5 - The base case is technically correct despite the irrelevant swap. However, the omitted `h'` in the inductive hypothesis significantly weakens the prediction and makes it much less useful.
>
> **Output:** [4, 2]
> ```

---

> ### Author Response · Authors · 2024-11-19
> **Q5 Autorater summary**
>
> The example above shows that using a large LLM can parse and interpret the semantics of the two predictions,
> allowing it to score the semantics better than what would be allowed by a lexical similarity such as BLEU.
> Of course, there are cases where the LLM is incorrect, however previous work (e.g. related work in [2]) has shown that this approach
> is still quite effective.
>
> We ran this prompt with 1383 prediction comparisons for transitions from the LeanDojo Novel Premises benchmark.
> We ran it twice for each example, where we swap the
> order of the predictions to remove ordering bias in the LLM. We then average the scores over the two orderings.
> For this, the 95\% CI for the COMBINED model was (2.7, 2.9) while the 95\% CI for the SEPARATE model was (2.0, 2.2)
> This shows a significant increase in the LLM scores for the COMBINED vs SEPARATE model, where it is prompted to
> score for semantics rather than syntax.
>
> We hope that this gives you some more confidence in the predictions from our transition model,
> and that it can capture the semantics of the resulting environment state.
>
> References:
>
> [1] Husein et al., Large language models for code completion: A systematic literature
> review, https://doi.org/10.1016/j.csi.2024.103917.
>
> [2] Vu et al, Foundational Autoraters:
> Taming Large Language Models for
> Better Automatic Evaluation, https://arxiv.org/pdf/2407.10817

---

> ### Author Response · Authors · 2024-11-19
>
> # Q.6
> *In lines 408-418, I do not see the quoted values in the referred tables: "∼36% relative improvement (Table 1)" and "∼6–9% relative improvements (Table 7)". It is not clear if these refer to the tables in the respective papers.*
>
> Thank you for pointing this out, these are references are to tables in the respective papers, which we will make explicit in the revision.
>
> # Q.7
>
> *I have concerns about the training dataset used for the Transition Model. Specifically, in lines 220-222, it is
> mentioned that "We obtain the dataset D from a vanilla ReProver attempt on miniF2F-valid, which results in 498,236
> transitions, which we split randomly into 95\% training, 5\% testing." i.e., 3D-Prover uses a transition model trained
> from miniF2F-valid transitions. In that case, doesn't that mean that in Table 2, the proof search results for
> miniF2F-valid, are trained and evaluated on the same dataset?*
>
> You are correct to observe that our miniF2F-valid results use a transition model trained on one attempt of the same
> dataset,
> as we specify in line 421. We note that this is a common paradigm, with many previous proof search methods (e.g. [1, 2, 3] )
> using previous attempts to improve subsequent approaches over the same dataset. This is referred to
> as, for example, *Expert Iteration* in [2], or a *transductive* setup in [1], where we quote from section 6.4 in [1]:
>
> **This protocol is also sensible, as allowing the model to learn from a failed proof-search can lead to more focused exploration on the
> next attempt, proving more statements overall than a model that would not be trained online.**
>
> As they discuss, this is a reasonable paradigm, analogous to online learning in an RL setup,
> where the model learns from previous proof attempts to improve its subsequent performance.
> We will mention this setup in our revision, to make it clear.
>
> *And for this reason, the results for miniF2F-valid in
> Table 2 are significantly better than those for miniF2F-test? Further, the improvement in pass@1 values on the
> miniF2F-test is not that significant. For K=8, there is a 22.4\% to
> 24.4\% i.e. a 2\% increase in the number of proofs found. And this improvement goes down to 0.4\% (27.8\% to 28.2\%) for
> K=32.*
>
> It is not unexpected that when the transition model is trained over the same proof attempts it performs better,
> which as above is a reasonable online learning setup used to evaluate proof search methods.
> Although the improvement is less for miniF2F-test, it still shows the effectiveness of our approach in the
> more difficult scenario of unseen proofs.
>
> As we discuss from line 403, it is reasonable to expect tree search to improve
> performance by a relatively small degree, when compared to an improved tactic generator such as DeepSeek-Prover or
> InternLM-Math. When comparing to other search methods, we find that our approach is competitive.
> For example, the MCTS based search algorithm in DeepSeek-Prover-V1.5 (Figure 5 in [1]), over miniF2F-test,
> increases performance from 58.4% to 59.6% for the 4-pass setting, and 60.2% to 62.7% for the 16 pass setting (relative
> improvements of 2.1% and 4.2% respectively).
> The proofsize objective value function from [2]
> increases performance (on miniF2F-valid, Table 1 in [2]) from 28.4% to 28.5% for the 1-pass setting,
> and 33.6% to 35.5% for the 8 pass setting (relative improvements of 0.04% and 5.7% respectively).
> They also train for 2 iterations (whereas we train for 1), with each iteration taking around 20,000 GPU hours (we take
> approximately 100).
>
> Our results in Table 2 show strong improvements for deeper proof settings, which are more difficult to achieve.
> Our experiment on the larger LeanDojo dataset (G.2) demonstrate this further, where we find more (and deeper)
> proofs in comparison to no filtering, and the Top-K baseline.
>
>
> References:
>
> [1] Lample et al., HyperTree Proof Search for Neural Theorem Proving, https://arxiv.org/pdf/2205.11491
>
> [2] Polu et al., Formal Mathematics Statement Curriculum Learning, https://openreview.net/pdf?id=-P7G-8dmSh4
>
> [3] Bansal et al., HOList: An Environment for Machine Learning of Higher Order Theorem Proving, https://arxiv.org/pdf/1904.03241

---

> ### Author Response · Authors · 2024-11-19
>
> # Q.8
>
> *One of the most concerning weaknesses of this paper is that it does not compare the ReProver (for tactic prediction) +
> 3D-Prover (for tactic filtering) framework against any state-of-the-art method like DeepSeek-Prover, InternLM-Math, etc.
> The filtering method is only compared with two baseline tactic filtering mechanisms i.e., Random filtering and top-k
> filtering. But the paper does not show whether the effect of this filtering is significant enough, such that it
> outperforms other state-of-the-arts that do not use filtering or use some kind of state-of-the-art filtering.*
>
> We emphasise that our approach serves to improve search given an
> arbitrary tactic generator, so it can be applied to new models as they are developed (which
> is important, given the rapid release of new and improved tactic generators).
> It should therefore be compared to other methods improving search, rather than to the tactic generator itself, which we discuss above.
> We are also not aware of any state-of-the-art filtering approach in this area to compare to.
>
> Due to resource constraints, we were only able to evaluate our framework using the smaller ReProver model as the base
> tactic generator (which we mention in line 348), where we improved the performance without modifying the base model.
> Current state-of-the-art methods are significantly larger, and it was not feasible for us to run the scale of our
> experiments on our hardware using these as a tactic generator (ReProver is approximately 300M parameters compared to the
> 7B of DeepSeek-Prover or InternLM).
>
> References:
>
> [1] Xin et al., DeepSeek-Prover-V1.5: Harnessing Proof Assistant Feedback for Reinforcement Learning and Monte-Carlo Tree Search, https://arxiv.org/pdf/2408.08152
>
> [2] Polu et al., Formal Mathematics Statement Curriculum Learning, https://openreview.net/pdf?id=-P7G-8dmSh4

---

### Official Review · Reviewer_HiN1 · 2024-11-09

**Soundness:** 3
**Presentation:** 3
**Contribution:** 3
**Rating:** 6
**Confidence:** 3

**Summary:**

The search space of proofs grows exponentially with the depth of the proof but a number of branches in this space capture sequences that could be semantically similar or lead to errors. Pruning this search space is, thus, critical. The paper looks at using synthetic data from proof attempts to accomplish this pruning. Semantically-aware tactic representations are generated to capture the effect on proving environment and likelihood of success. DPPs are used to select semantically diverse and high quality tactics using these representations.  The developed approach is called Diversity Driven Determinantal Point Process Prover (3D-Prover). The approach is evaluated on miniF2F-valid and miniF2f-test benchmarks by augmenting reProver LLM.

**Strengths:**

Many generated proof paths are equivalent modulo variable renaming and other semantics-preserving transformations (such as variable renaming). Authors have found that 75% of the proof paths lead to execution error and thus, it is critical to factor them into exploration. Past work has considered sparse binary signal from the proof status of a goal, intrinsic reward for exploration for new nodes, etc but these do not factor in error likelihood, error messages and execution time. Further, addition of new nodes does not factor in node similarity.

3D-Prover can augment proof search by filtering candidate tactics to generate diverse and high quality subsets. It is able to filter tactics based on their likely outcome.

The utility of the transition model representations is demonstrated using ablation study where the transition model Encoder is replaced by an Autoencoder of the same size. Augmenting the ReProver LLM on the standard miniF2F benchmark, the paper reports an improvement in the overall proof success rate.

**Weaknesses:**

Please see questions for some of the concerns of the reviewer. The reviewer is happy to raise the score if the concerns are addressed.

**Questions:**

For representing the transitions, the paper adopts a specific architecture - encode the goal and tactics followed by a decoder to get the outcome (next goals or error) and a predictor to determine whether the transition led to an error and the time taken. There is no intuition provided to why this is a better architecture. Why note just have a decoder predict the entire tuple? What other architectures were considered and why is the current one most promising? Even if there is no experimental evaluation of other baseline, it would be good to include discussion based on whatever experimentation was done to select this architecture.

When splitting the 500K transitions into train and test, how did the authors ensure that it is not the case that the same goal, tactics, next-goal triplet in the test set does not show up in another proof and has been included in the training set. Since the transitions are being collected across several proofs, how is the train/test partitioning ensured to be non-overlapping?

Could you include not just top-4 but also top-k (k= 1 to 4 or 5) in the Table 1?

---

> ### Author Response · Authors · 2024-11-19
>
> Thank you for the detailed and helpful comments. We appreciate your willingness
> to increase the score if we can address your concerns, which we believe are all reasonable and achievable with
> some additional discussion and minor updates. We will address your concerns in order below.
>
> # Q.1
>
> *For representing the transitions, the paper adopts a specific architecture - encode the goal and tactics followed by a
> decoder to get the outcome (next goals or error) and a predictor to determine whether the transition led to an error and
> the time taken. There is no intuition provided to why this is a better architecture. Why note just have a decoder
> predict the entire tuple? What other architectures were considered and why is the current one most promising? Even if
> there is no experimental evaluation of other baseline, it would be good to include discussion based on whatever
> experimentation was done to select this architecture.*
>
> We agree that an expanded discussion motivating our transition model architecture will help with the clarity of the
> paper. Our general comment G.1 provides the intuition and motivation of our architecture, however we will expand this to
> address your specific questions.
>
> We don't have the Decoder predict the entire tuple for two reasons. Firstly, we require our time
> prediction to be real valued, and the status prediction to be in [0,1].
> This is an unnatural prediction task for the decoder, which would require us to e.g. discretise the time into several buckets,
> complicating the architecture further.
> Secondly, we wish to have a fast and efficient Predictor, as it is used in the 3D-Prover algorithm to evaluate tactics.
> Using a small MLP for the predictor is therefore useful for speeding up 3D-Prover, as we can discard the much
> larger Decoder (only needing the Encoder and Predictor for 3D-Prover). As discussed in G.1,
> we will update the paper to clarify these points, as we agree it adds important context to motivate the architecture.
>
> We hope that this clarifies the intuition and motivation behind our architecture, but please
> let us know if you have any further questions.
>
> # Q.2
>
> *When splitting the 500K transitions into train and test, how did the authors ensure that it is not the case that the same goal,
> tactics, next-goal triplet in the test set does not show up in another proof and has been included in the training set.
> Since the transitions are being collected across several proofs, how is the train/test partitioning ensured to be non-overlapping?*
>
> Addressing the concern of overlapping train and test sets,
> we note that every goal state (within and between proofs) is unique.
> The search tree is implemented to ensure that all nodes for a given proof have unique goal states,
> with tactics for a given node being all unique (so there is no overlap within proofs).
> The goal state includes a unique identifier for the proof attempt it belongs to (preventing overlap between proofs).
> These factors prevent any overlapping triplets within and between proofs,
> as we only use transitions from a single proof attempt per goal.
>
> Your comment did however raise the question: If we ignore the unique identifier,
> are there overlapping tuples between proofs as you suggest (where the goal, tactic and response are identical for different proof attempts)?
> In this case, the model might ignore the identifier and learn to predict the response based on a previously seen example.
> To address this concern, we examined our dataset and found 136 of these overlaps out of the 498,236 transitions.
> Although such a small overlap would not impact our results in any noticeable way,
> we will update our data processing to detect and remove these instances.
>
> We appreciate your identification of this, which will help improve the quality of our dataset.
>
> # Q.3
> *Could you include not just top-4 but also top-k (k= 1 to 4 or 5) in the Table 1?*
>
> We assume you refer to Table 8 in Appendix A1, for which we will add the Pass@k results for k=1 to 4 in the revision.
>
> For reference, the Pass@k for our experiments are in the table below, where we report the Pass@k based on the order of
> execution for each run:
>
>
>
> |        | 3D-Prover (K=8) | 3D-Prover (K=16) | 3D-Prover (K=32) | Random (K=8) | Random (K=16) | Random (K=32) |
> |--------|-----------------|------------------|------------------|--------------|---------------|---------------|
> | Pass@1 | 24.9%           | 27.8%            | 28.6%            | 18.0%        | 21.2%         | 28.1%         |
> | Pass@2 | 26.1%           | 29.4%            | 29.0%            | 22.9%        | 28.6%         | 29.0%         |
> | Pass@3 | 26.5%           | 29.8%            | 29.8%            | 24.9%        | 29.4%         | 29.8%         |
> | Pass@4 | 28.6%           | 31.0%            | 29.8%            | 25.7%        | 30.2%         | 29.8%         |

---

> > ### Comment · Reviewer_HiN1 · 2024-11-25
> > **Thank you**
> >
> > Thank you for your response and the analysis.

---

> > > ### Author Response · Authors · 2024-11-26
> > >
> > > No problem. Was our response able to address your concerns, or is there is anything else you would like addressed to raise your score?

---

### Author Response · Authors · 2024-11-19
**General Comment**

We would like to thank all the reviewers for their detailed and constructive feedback.
We appreciate the time and effort you have put into reviewing our paper,
and we are grateful for the opportunity to address your comments.

# G.1

Based on the comments of reviewer HiN1 and 1vwm, we agree that the motivation and intuition behind our transition model architecture in Section 2 could be improved.
We understand that this may have caused some confusion, so we make a general comment
here to clarify (which we will adapt and include in our revision).

The primary motivation of the architecture shown in Figure 2 is to generate
appropriate feature vectors to enable the application of DPP in the 3D-Prover algorithm.
DPP requires a vector encoding the attribute which we want to sample diversely.
By generating vectors which reflect the impact of tactics on the environment, our transition model architecture hence
enables DPP to select tactics based on the diversity of their outcome.

Using the Encoder first, before the Decoder/Predictor, is done to enable the learning of these representation vectors.
By bottle-necking the tactic to this encoding, the Encoder must learn an effective representation to ensure that the
Decoder and Predictor have enough information to determine the subsequent effect of the tactic on the environment.
If we used a Decoder only architecture, this would not provide us with the tactic representation vector for DPP in the 3D-Prover algorithm.
The ALL TOKENS and NO TACTIC baselines provide some alternative architectures for comparison, showing how
effective the Encoder is at generating useful representations for these predictions.

Following training, 3D-Prover discards the Decoder, using only the Encoder and Predictor to evaluate candidate tactics
as outlined in Algorithm 1, so we wish to make the Predictor as small and efficient as possible (hence the small MLP architecture for the Predictor).

It is true that alternative architectures might improve upon this, however we demonstrate ours is effective both for
predicting the environment outcome, and for generating
useful representations which can improve proof search (as we show in the Autoencoder comparison in 3.3.2 and Appendix
A.3). The primary contribution of the paper is in the search augmentation, so an improved architecture here
would only serve to further improve upon that.

# G.2

We ran an additional experiment on the LeanDojo Novel
Premises benchmark testing 3D-Prover on a larger dataset.
This dataset has 2000 proofs in comparison to the 244 from miniF2F-valid and miniF2F-test,
allowing us to evaluate the performance of 3D-Prover on a larger scale.

Following the same methodology, we trained a transition model from a single ReProver attempt,
before evaluating 3D-Prover, using K=32 for the filtering. We compare to the model with no filtering, and top-K=32.

Additionally, we examine the distribution of proof lengths found from this experiment.
To account for different proofs of the same goal, we adjust proof lengths to be the shortest found from any attempt (
e.g. if 3D-Prover finds a proof of length 10, which was found in 3 steps by No Filtering, we count it as length 3).
Hence, all proof lengths reported are the shortest found by any method.

We report the number of proofs found by each approach, organised by the proof length in the below table.

| Proof Length | 3D-Prover (K=32) | Top-K (K=32) | No Filtering (K=64) |
|--------------|------------------|--------------|---------------------|
| 1            | 236              | 233          | **237**             |
| 2            | 167              | 162          | **174**             |
| 3            | **134**          | 126          | 131                 |
| 4            | **60**           | **60**       | 54                  |
| 5            | **40**           | 39           | 24                  |
| 6            | **7**            | 6            | 2                   |
| 7            | **2**            | 0            | 0                   |
| Total        | **646**          | 626          | 622                 |

To summarise the results of this experiment, we found a relative improvement of 3.2\% over top-K=32, and a 3.9\%
relative improvement over no filtering in terms of the number of proofs found.
We see that 3D-Prover finds deeper proofs, while maintaining a high proof success
rate for shallower proofs, unlike Top-K. The no filtering approach, as expected, finds the most shallow proofs,
however quickly drops off in performance for deeper proofs.
We also note that 3D-Prover found the 2 longest proofs of length 7, with neither baseline finding any.

This gives some additional confidence in the benefits of our approach for a larger dataset,
with the improvement over no filtering addressing some concerns of reviewer 1vwm.

---

### Author Response · Authors · 2024-11-22
**First Revision Details**

We have submitted our first revision, where we have made changes to reflect the suggestions of the reviewers. These suggestions have greatly improved the clarity and flow of the paper, and helped strengthen our results. We have used blue font to highlight the major updates to our manuscript, making it easier to compare. If the reviewers have any additional comments or concerns please let us know so we can address them before the end of the discussion period.

A summary of the changes is as follows:

- We have added results from No Filtering (Table 2,4,5,6,7) as an additional baseline, to make the results more compelling as suggested by reviewer 1vwm (Q.1 and Q.2). In all cases, our approach leads to an improvement.
- We move the section introducing DPPs to the Introduction (1.2), helping us better motivate and discuss our transition model in 2.1 (addressing HiN1 Q.1, 1vwm Q.4). We have also expanded the motivation of DPPs, in line with comments from reviewer xeEr (Q.1).
- We have moved our preliminaries section to Appendix A.1, and simplified the notation and introduction of our transition model in section 2.1 (1vwm Q.3).
- We include the LeanDojo results from comment G.2 as an appendix (A.6), where we show the improvement of our approach over a larger dataset.
- We include Pass@k up to k=4 in Appendix A.2, addressing HiN1 Q.3.
- We clarify the references from 408-418 (1vwm Q.6)
- We explicitly discuss the filtering performance of hyperparameters (xeEr Q.2, lines 419-421)
- We note the online learning setup of miniF2F-valid, addressing 1vwm Q.7 (lines 422 - 424)
- We discuss the motivation behind preferring faster tactics (xeEr Q.3, lines 517-521)

---

### Author Response · Authors · 2024-12-04
**Summary of Reviews and Author Responses**

We are extremely grateful to all reviewers for their insightful comments and engagement with our work.
Our proposed approach was recognised as a fresh (hTv9), innovative (1vwm), well-motivated and practical (xeEr)
perspective on tree search in theorem proving.
Our evaluation demonstrated our approach to be effective (xeEr), with strong and impressive (1vwm) results,
where we established an improvement upon ReProver on miniF2F-valid and miniF2F-test without modifying the underlying
model (HiN1, xeEr, 1vwm, hTv9).
Based on reviewer feedback, we strengthened our results further with comparisons to a no filtering setup (1vwm), as well as
evaluating over a larger dataset (Appendix A.6, comment G.2).

We thank reviewer HiN1 and xeEr for their positive support of the paper,
with all of their concerns addressed and incorporated into the current revision.
We are grateful to reviewer 1vwm, whose suggestion of a comparison to a no filtering setup has
helped to further strengthen our results.
We have included detailed responses to the remaining suggestions/concerns of 1vwm in this discussion,
which has helped improve our revision as detailed below.
We finally thank reviewer hTv9 for their comments and active engagement in the discussion.
We believe we have addressed their major concern regarding the comparative computational cost of our approach, with
an additional experiment showing improvements when filtering time is included in the environment budget.

---

### Meta-Review · Area_Chair_rHeZ · 2024-12-25

**Metareview:**

This paper concerns pruning proof search of neural theorem proving and proposes 3D-Prover, which is built on top of the previous work ReProver and consists of a representation learning component capturing the transition semantic of theorem proving, and a filtering mechansim for proof tactics using Determinantal Point Processes (DPPs). DPPs are used to select semantically diverse and high quality tactics. The experimental evaluation shows that 3D-Prover outperforms baselines like Top-K and random filtering. Using DPPs to filter proof tactics in the context of neural theorem proving is novel, however, its effectiveness is not significant. Reviewers and the AC share concerns regarding the evaulation setup (e.g., filtering vs no-filtering) and effectiveness (e.g., marginal improvement compared to simple baselines).

**Additional Comments On Reviewer Discussion:**

During the rebuttal period, the authors shared examples and new results upon suggestions from reviewers (hTv9,1vwm,HiN1). The new information does help to resolve some concerns, but the main concern about the marginal improvement still remains.

---

### Decision · Program_Chairs · 2025-01-22

Reject